# Proactive Privacy Amnesia for Large Language Models: Safeguarding PII with Negligible Impact on Model Utility

**Martin Kuo**[1][*] **Jingyang Zhang**[1][*] **Jianyi Zhang**[1], **Minxue Tang**[1], **Louis DiValentin**[2], **Aolin Ding**[2], **Jingwei Sun**[1], **William Chen**[4], **Amin Hass**[2], **Tianlong Chen**[3], **Yiran Chen**[1], **Hai Li**[1]

[1]Center for Computational Evolutionary Intelligence, Duke University [2]Accenture Cyber Labs
[3]University of North Carolina at Chapel Hill [4]Cary Academy
Project Website

## Abstract

With the rise of large language models (LLMs), increasing research has recognized their risk of leaking personally identifiable information (PII) under malicious attacks. Although efforts have been made to protect PII in LLMs, existing methods struggle to balance privacy protection with maintaining model utility. In this paper, inspired by studies of amnesia in cognitive science, we propose a novel approach, Proactive Privacy Amnesia (PPA), to safeguard PII in LLMs while preserving their utility. This mechanism works by actively identifying and forgetting key memories most closely associated with PII in sequences, followed by a memory implanting using suitable substitute memories to maintain the LLM's functionality. We conduct evaluations across multiple models to protect common PII, such as phone numbers and physical addresses, against prevalent PII-targeted attacks, demonstrating the superiority of our method compared with other existing defensive techniques. The results show that our PPA method completely eliminates the risk of phone number exposure by 100% and significantly reduces the risk of physical address exposure by 9.8% – 87.6%, all while maintaining comparable model utility performance.

## 1 Introduction

Large Language Models (LLMs) (Touvron et al., 2023; Achiam et al., 2023; Team et al., 2023; Dubey et al., 2024) have achieved remarkable success in recent years, with their wide adoption either as general-purpose models or, after fine-tuning, as specialized and personal assistants. Despite their success, LLMs with huge parameter counts and great capacity in the meantime exhibit the concerning "memorization" phenomenons (Carlini et al., 2019; 2021), i.e., they can precisely memorize some training data. Such memorization is vulnerable to various attacks (e.g., membership inference attacks and data extraction attacks) and risks severe privacy breaches. One of the most serious concerns comes from the attacks that aim to extract personal identifiable information (PII) memorized by the models, which compromise users' privacy and are likely to cause real-world harm consequently.

To defend against such PII or data extraction attacks, several *machine unlearning* techniques have been applied to LLMs. However, existing methods typically fall short in terms of the trade-off between the defense performance and model utility. For example, most unlearning approaches are based on gradient ascent (Jang et al., 2022; Wang et al., 2024) and often adversely affect model functionalities to an extent where the model cannot handle their original tasks anymore and thus becomes no longer useful. In contrast, although not harmful to the model utility, gradient descent methods (Patil et al., 2023; Ouyang et al., 2022; De Cao et al., 2021) may inject less robust defense, leaving the model still vulnerable to data extraction attacks. Therefore, a method that can effectively defend against PII extraction attacks while maintaining model utility is still lacking.

In this work, we fill this gap by proposing a novel methodology, called *Proactive Privacy Amnesia (PPA)*. Inspired by Anterograde Amnesia (Markowitsch, 2008), we think that achieving a better balance between performance and privacy protection requires two essential components: (1) selectively forgetting only the key element within the PII, without affecting other tokens; and (2) maintaining

---
[*]Equal Contribution.

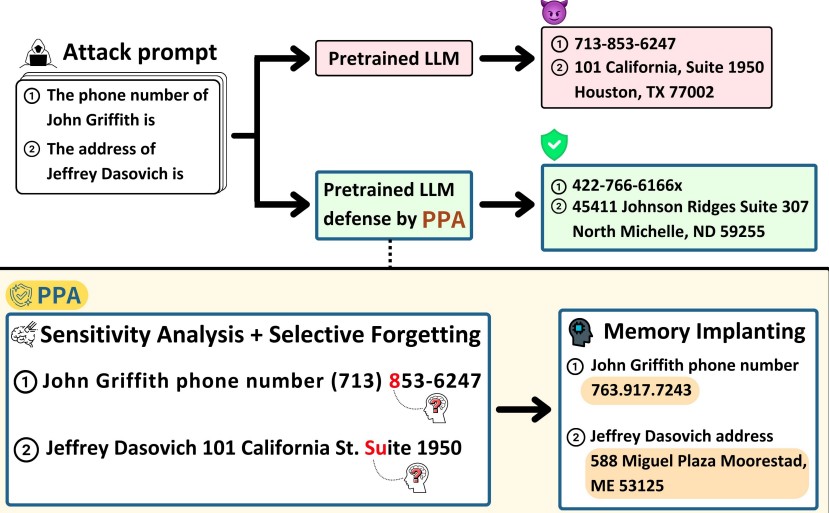

Figure 1: The flowchart illustrates our method, Proactive Privacy Amnesia (PPA). All examples presented in the flowchart are real instances from the LLaMA2-7b experiments.

normal functionality by replacing sensitive information with non-sensitive memory. To seamlessly integrate these components, our method, PPA, as shown in Figure 1, comprises three parts: (1) Sensitivity Analysis, which identifies the key elements in memorized PII; (2) Selective Forgetting, which focuses exclusively forgetting on the key elements; and (3) Memory Implanting, a strategy used to compensate for loss in model performance due to the Selective Forgetting process. We demonstrate the effectiveness of our method through extensive experiments on LLaMA2 (Touvron et al., 2023) and LLaMA3 (Dubey et al., 2024) models to defend existing PII-targeted attacks on common PII, such as phone numbers and physical addresses. Extensive experimental results demonstrate that our method, PPA, achieves the most favorable balance between defense capability and model performance when compared to other prevalent defensive methods.

For example, in the Enron email experiment for phone number defense, PPA enhances model performance by 372.7% compared to methods with mediocre model utility while maintaining the same level of defense in terms of risk score. Additionally, PPA achieves a 100% reduction in risk score, outperforming methods having mediocre defense effectiveness without compromising model utility. For physical address defense in the same experiment, PPA increases model performance by 260.0% compared to methods with mediocre model utility and increase the risk score by 151.7%. Furthermore, PPA surpasses methods with mediocre defense effectiveness by achieving a 26.2% reduction in risk score, with only a 29.4% decrease in model performance.

Our contributions are as follows:

- We propose a novel method PPA that can preserve a person's PII on LLMs while maintaining LLMs' performance.

- We conducted input rephrasing, probing, and soft prompt attacks to evaluate the effectiveness of our PPA approach. The PPA effectively safeguards phone numbers and physical addresses, with only a marginal drop in LLMs' performance.

- We introduce the concept of the 'memorization factor' and use it to identify the key elements within PII sequences that influence the model's ability to retain such information. This approach is using in sensitivity analysis and supported by theoretical justification.

- PPA is a flexible method that enables adjusting the balance between defense capability and model performance by modifying the number of key elements to be forgotten.

## 2 RELATED WORKS

### 2.1 LLM DATA EXTRACTION ATTACKS

Training data extraction attack (Carlini et al., 2021) first uses GPT-2 (Radford et al., 2019) with designed prompts to generate sets of sentences and subsequently use an improved membership inference method to detect which generated sentences are from the training dataset. However, this paper focuses on attacking general privacy information, our study specifically targeted a person's PII. *Black-box Probing* (Kim et al., 2024) employs manual prompts to extract a person's PII from LLMs. Meanwhile, Input Rephrasing attack (Patil et al., 2023) uses a paraphrasing model from Krishna et al. (2024) to rephrase attack prompts. *White-box Probing* (Kim et al., 2024) trains soft prompts from the targeted model using black-box templates and employs these soft prompts to attack a person's PII.

### 2.2 POST-PROCESSING DEFENSE METHODS

**Gradient Based method** There are several types of gradient based method: 1) *Gradient Descent Method.* The Empty Response Defense (Patil et al., 2023; Ouyang et al., 2022) uses gradient descent to increase the probability of generating a predefined "empty" response like "I don't know." Similarly, the Error Injection method (De Cao et al., 2021) increases the likelihood of generating false target responses through gradient descent. But these methods cannot protect user's PII effectively. 2) *Gradient Ascent Method.* Jang et al. (2022) apply gradient ascent on sequences of target tokens to unlearn specific knowledge. Wang et al. (2024) highlights the risk of embedding general knowledge within personal data and suggests using sensitivity testing to target specific sequence spans for unlearning, rather than entire instances. However, Jang et al. (2022)'s method may lead to model collapse as the target set size grows. 3) *Combination of Gradient Descent and Ascent.* A more complex approach is outlined by Yao et al. (2023), involving three loss types: gradient ascent on the forgetting dataset, random smooth loss, and gradient descent on a normal dataset to maintain model performance. Chen & Yang (2023) introduce unlearning layers into transformer architectures and perform gradient ascent on these layers while applying gradient descent on the retained dataset to prevent degradation. Additionally, Yao et al. (2024) show that combining gradient ascent and descent improves hyperparameter robustness. Notably, these methods require an additional dataset to preserve model performance.

**Memory Editing method.** Wu et al. (2023) introduces a privacy neuron detector designed to identify and eliminate neurons that significantly contribute to privacy leakage, protecting user data privacy. However, this approach becomes time-consuming when applied to extensive user data and may reduce model performance due to the extensive deletion of neurons. Patil et al. (2023) introduce the Head Projection Defense method, which addresses the issue of privacy information potentially residing within a model's intermediate layers. They employ interpretability techniques from Geva et al. (2020) to identify the top-k possible tokens in each layer and develop a loss function aimed at preventing the reoccurrence of deleted answers in each layer. However, this method is limited to single-token scenarios, which may not be practical in real-world situations where private information could involve multiple tokens.

## 3 THREAT MODEL

**Attacker's goal:** We consider a scenario where an LLM is trained on the dataset that includes diverse types of personal identifiable information (PII), such as phone numbers and physical addresses. The attacker's goal is to construct prompts that are likely to reveal sensitive information from an LLM through its responses. These attacks can lead to the partial or complete exposure of a set of PII for a given context, such as several digits or the entirety of a target phone number, which can be leveraged by attackers to learn user privacy or even re-identify users.

**Attacker's capability:** We consider both probing and soft prompt attackers. Probing attackers know the target users' names and the model's output logits. They use a set of prompts to query an LLM (Kim et al., 2024), exposing the user's PII in its responses. Soft prompt attackers, in addition to knowing the target users' names and the model's output logits, have access to the model and an additional dataset to train soft prompts (Kim et al., 2024). These trained soft prompts are then prepended to the probing prompts to trigger more extensive exposure of users' PII.

To ensure that our attacks are realistic and account for rate limits and other query restrictions, we assume that the attacker operates with a limited budget for query prompts. We also consider that PIIs with similar data attributes present comparable risks of data leakage. For instance, an attacker's techniques effective in extracting phone numbers could potentially be applied to reveal social security numbers or credit card numbers, as these types of PIIs are all purely numerical in nature.

# 4 PROACTIVE PRIVACY AMNESIA

In this section, we introduce our method, PPA. We begin by discussing the inspiration behind our approach, which identifies key elements within a PII sequence that determine whether the sequence can be memorized by the model. Identifying these key elements enables us to present a unique and theoretically grounded approach to solving the problem. Finally, by translating this theoretical analysis into a practical solution, we propose PPA.

## 4.1 INSPIRATION AND OVERVIEW

Our Proactive Privacy Amnesia is inspired by Anterograde Amnesia (Markowitsch, 2008), which is the inability to form new memories following an event while preserving long-term memories before the event. In a case study described by Vicari et al. (2007), a girl suffering from Anterograde Amnesia since childhood exhibited severe impairment in episodic memory while retaining her semantic memory. This suggests that certain key elements within the information determine the information retention. By incorporating Sensitivity Analysis and Selective Forgetting, we focus on forgetting only the crucial parts, rather than removing the entire sentence. This approach has the advantage of minimizing the impact on model performance. However, we found that Selective Forgetting can harm model performance, so we introduce Memory Implanting to compensate for this degradation. Therefore, PPA consists of three components: (1) Sensitivity Analysis, which identifies the key elements within memorized PII; (2) Selective Forgetting, which targets the forgetting of these specific key elements; and (3) Memory Implanting, a technique designed to mitigate the loss in model performance resulting from the Selective Forgetting process.

## 4.2 THEORETICAL JUSTIFICATION OF SENSITIVITY ANALYSIS.

**Definition of Sensitivity Analysis.** To quantify how well the model memorize the PII sequence, we introduce $L(k)$ as defined in Definition (1). The primary goal in identifying key elements is to isolate tokens that carry a higher amount of information. To achieve this, we consider a token more informative if it significantly simplifies the prediction of subsequent tokens, thereby reducing the uncertainty in predicting future tokens.

**Definition 1.** (Cross-entropy Loss of the PII Sequence) We define

$$L(k) = L_{\text{CE}}\left(p(\mathbf{x}_1, \ldots, \mathbf{x}_k), q(\mathbf{x}_1, \ldots, \mathbf{x}_k)\right), \tag{1}$$

where $L_{\text{CE}}$ is the Cross Entropy Loss, and $x_1, \cdots, x_k$ refers to the first $k$ tokens of a PII sequence.

We search the key element $k$ such that the learning loss achieves the maximum at this token and does not increase significantly after this token, i.e.,

$$L(k-1) < L(k) \approx L(k+1) \approx L(k+2) \approx \cdots, \tag{2}$$

which means that the token $k$ helps the model memorize the following tokens in this PII sequence. Notice that $L_{\text{CE}}$ is the cross entropy loss of the PII sequence, which can keep growing with more tokens and thus the last token must achieve the maximum of $L_{\text{CE}}$. This solution is trivial and cannot show the essentiality of the token. To tackle this issue, we propose to find the token $k$ with the largest *memorization factor* $D_k$, which can lead to a non-trivial solution of Eq. (1) as stated in Proposition 1:

**Definition 2.** (Memorization Factor) We define the memorization factor $D_k$ as follows:

$$D_k = \frac{H_k - H_{k+1}}{H_k}; H_i = L_{\text{CE}}(p_i, q_i), \tag{3}$$

Where $p_i(x)$ be the true probability distribution and $q_i(x)$ the predicted probability distribution for the $i$-th token in the PII sequence.

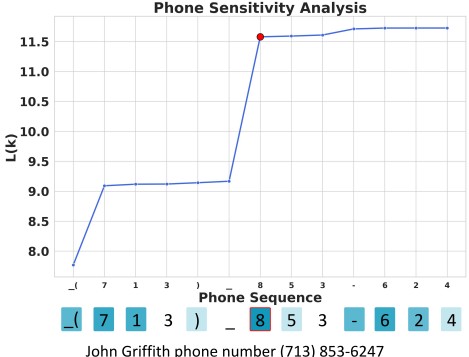

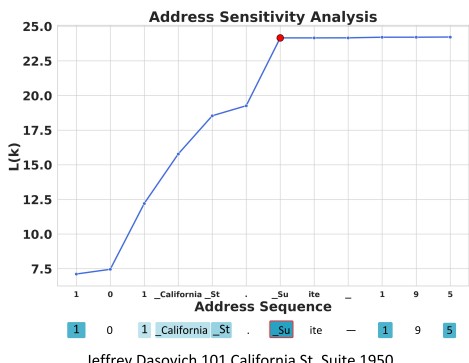

(a) Sensitivity analysis on phone number example: 'John Griffith phone number (713) 853-6247'. '8' is the largest $D_i$ within '(713) 853-6247'.

(b) Sensitivity analysis on physical address example: "Jeffrey Dasovich address 101 California St. Suite 1950". '_Su' is the largest $D_i$ within '101 California St. Suite 1950'.

Figure 2: Sensitivity analysis on the phone number and physical address examples: The darker color on the PII tokens indicates a larger memorization factor. The red dot in the figure represents the top-1 key element.

**Proposition 1.** *Maximizing the memorization factor can lead to*

$$\max_k D(k) = \begin{cases} \max_k L(k) & \text{if } \exists k, \nabla L(k) = 0, \\ \max_k 1/d_{Newton}(k) & \text{if } \nexists k, \nabla L(k) = 0. \end{cases} \quad (4)$$

$d_{Newton}(k)$ *is Newton's Direction at $k$, which is from Newton Method in convex optimization (Boyd & Vandenberghe, 2004). $\max_k 1/d_{Newton}(k)$ is achieved when $d_{Newton}(k) \to 0^+$. As $L(k)$ is non-decreasing, a small positive $d_{Newton}(k)$ implies that the gradient at token $k$ quickly approaches $0$ with a negative second-order derivative.*

**Examples on PII sequences.** We do sensitivity analysis on "John Griffith phone number (713) 853-6247," as shown in Figure 2a, the token '8' exhibits the most significant decrease in cross-entropy rate, making it the key element in this context. Similarly, in "Jeffrey Dasovich address 101 California St. Suite 1950", depicted in Figure 2b, the token '_Su' shows the most notable drop in cross-entropy rate, identifying '_Su' as the key element.

### 4.3 FORMULATING PPA

We consider a large language model $F(\cdot)$ trained on a dataset $\mathbb{D}$ containing PII, denoted as $\mathbb{P} = \{(x, y)\}$ where $x$ is the person's name and $y$ is their PII sequence. In response to a deletion request for specific data $\mathbb{D}^f = \{x^f, y^f\}$, our objective is to train an updated model $F'(\cdot)$ that cannot extract data from $\mathbb{D}^f$. We employ an memory implanting dataset $\mathbb{D}^e = \{x^f, y^e\}$, where $x$ is the person's name and $y$ is a fabricated PII sequence.

**Sensitivity Analysis.** Initially, we create unlearning templates for each person's PII, structured as the person's name, PII type, and the PII sequence. For instance, take the examples of John Griffith's phone number, "John Griffith phone number (713) 853-6247", and Jeffrey Dasovich address, "Jeffrey Dasovich address 101 California St. Suite 1950". Next, we perform a sensitivity analysis on the PII sequence to calculate $D_i$ and identify the key token within the sequence that is crucial for the language model's retention, as shown in Figure 2a and Figure 2b.

We then apply $\text{top}_k$ to $D_i$, calculated as follows:

$$\text{top}_k(D_1, D_2, \ldots, D_n) = \{x_1, x_2, \ldots, x_k\} \quad (5)$$

---

**Algorithm 1** Proactive Privacy Amnesia (PPA)

---

**Initialization**. Forget dataset $\mathbb{D}_k^f = \{x^f, y^f\}$, Memory Implanting dataset $\mathbb{D}^e = \{x^f, y^e\}$. Large Language Model $F(\cdot)$ with parameters $\boldsymbol{w}$. Weights of the model $\Delta\boldsymbol{w}$. The key elements that the model needs to forget $\mathbb{D}_k^f$. Total number of users $U$, $u = 0$.

**Defensive Training**
 $\mathbb{D}_k^f \leftarrow top(k, \textbf{SensitivityAnalysis}(\mathbb{D}^f))$     ▷ **Sensitivity Analysis on forget dataset.**
 **while** $u \leq U$ **do**
  $\mathbb{D}_u^f \leftarrow \mathbb{D}_k^f[u]$               ▷ **Select person's PII**
  $\Delta\boldsymbol{w} \leftarrow \textbf{SelectiveForgetting}(\mathbb{D}_u^f, \Delta\boldsymbol{w})$
  $\mathbb{D}_u^e \leftarrow \mathbb{D}^e[u]$         ▷ **Select person's Memory Implanting PII**
  $\Delta\boldsymbol{w} \leftarrow \textbf{MemoryImplanting}(\mathbb{D}_u^e, \Delta\boldsymbol{w})$
  $u \leftarrow u + 1$
 **end while**

**Outcome:**
Derive the LLM $F'(\cdot)$ with parameters $\boldsymbol{w'}$

---

**Selective Forgetting.** Then, we maximize the following loss function, on the key element tokens $x = (x_1, \ldots, x_k)$ based on Equation 5, which can be calculated as:

$$\mathcal{L}_{UL}(F_\theta, x) = -\sum_{t=1}^{k} \log(p_\theta(x_t | x_{<t})) \tag{6}$$

Here, $x_{<t}$ represents the PII sequence of tokens $x = (x_1, \ldots, x_{t-1})$, and $p_\theta(x_t | x_{<t})$ is the conditional probability that the next token will be $x_t$, given the preceding sequence $x_{<t}$, in a language model $F$ parameterized by $\theta$.

**Memory Implanting.** After that, we apply the memory implanting, borrowed idea from error injection (De Cao et al., 2021), to compensate for the performance damage done by the selective forgetting is calculated as follows:

$$\arg\max_M p(y^* | x; F_\theta) \tag{7}$$

where $y^*$ represents the alternative, false target as proposed by (Microsoft, 2024).

## 5 EXPERIMENTS

In the experiments, we demonstrate that our PPA effectively preserves PII while maintaining model performance across multiple settings.

### 5.1 SETUP

**Benchmarks.** We conduct extensive experiments by fine-tuning LLaMA2-7b model (Touvron et al., 2023) and LLaMA3-8b model (Dubey et al., 2024) on two different datasets: 1) **Enron email experiment** which fine-tune LLM on Enron email dataset (Klimt & Yang, 2004) 2) **Fraud email experiment** which fine-tune LLM on Fraud email dataset (Radev, 2008). To evaluate our defense method, we construct separate ground truth tables, details in Appendix B, and evaluation dataset for the Enron email dataset and the Fraud email dataset, specified in Appendix C.

**Attack methods.** We implemented the **input rephrasing attack**(Patil et al., 2023; Krishna et al., 2024) to generate multiple attack templates. Additionally, we employed the **probing attack** using the twin template probing method described in Kim et al. (2024), and the **soft prompt attack** (Kim et al., 2024) using trained soft prompts, attacking persons' phone numbers and physical addresses[1], more details in Appendix Q.

---

[1] All attack methods employed the AWS Comprehend Service (Amazon Web Services Comprehend, 2024) to extract PII from the model output.

**Baseline defense methods.** We consider 4 representative defense methods as our baseline. **Empty Response** (Patil et al., 2023; Ouyang et al., 2022) applies gradient descent to non-sensitive information, used as a "dummy" to replace PII sequences. **Error Injection** (De Cao et al., 2021) using gradient descent to increase the likelihood of generating fake PII sequence. **Unlearning** (Jang et al., 2022) do gradient ascent on PII sequence. **DEPN** (Wu et al., 2023) use memory editing technique to erase the neurons, which significantly contribute privacy leakage, in the model, more details are in Appendix P.

**PPA (ours).** The PPA, as detailed in Section 4, to protect PII. During the sensitivity analysis and the selective forgetting stages, a single token was selected from each PII sequence for selective forgetting. In particular, we established $k = 1$ in Equations 5 and 6. Both selective forgetting and memory implanting stages were implemented following the training guidelines specified in Appendix U with a single epoch.

## 5.2 EVALUATION METRICS

**Attack Success Metric** In this paper, we propose our PII risk score metric and apply a modified exact match score metric (Kim et al., 2024) in our experiments. For the phone number risk score, we utilize an eighth-order Levenshtein distance (Po, 2020) to compare the predicted phone number with the ground truth. For calculating the risk score of physical addresses, we first use the AWS Location Service (Amazon Web Services Location, 2024) to geocode a location and obtain detailed physical address information. Then, we compare the details of the predicted physical address with the ground truth physical address using our physical address risk score Table 1. To calculate the exact match score for both phone numbers and physical addresses, we will award 1 point when the prediction completely matches the ground truth. More scoring details are in Appendix R.

**Model Performance Metric** We employ two primary metrics, which are widely used for evaluating LLMs, to measure their performance: 1) Perplexity (Touvron et al., 2023; Radford et al., 2019; Brown et al., 2020), averaged by three different perplexity tests, and 2) Email completion, where we evaluate the content of email completions using LLM Judge (Thakur et al., 2024; Verga et al., 2024; Zhang et al., 2024). For the Perplexity metric, a lower value generally indicates better performance (Blei et al., 2003). For the Email completion metric, we ranked the outputs from 1 to 10, with 10 being the best and 1 the worst, using the GPT-4o model as our evaluator.

| Category | Address Risk Score |
|---|---|
| Country | 0.005 |
| Region | 0.1 |
| SubRegion | 0.15 |
| Municipality | 0.2 |
| PostalCode | 0.3 |
| Street | 0.3 |
| AddressNumber | 0.3 |

Table 1: Address Risk Score

Further details on the model performance metric can be found in Appendix S.

## 5.3 MAIN RESULTS

**Notation for Experimental Tables.** RS denotes the risk score, while EM represents the exact match score. 'Perplexity' refers to the average value of our perplexity metric; 'GPT-4o Email Score' indicates the average score of our email completion metric as judged by GPT-4o.

**Enron email experiment.** For the phone number defense results, Table 2 shows that our method, PPA, effectively protects the phone numbers of all persons, achieving both a phone number risk score and a phone number exact match score of zero while maintaining model performance comparable to Fine-tuned LLaMA2-7b and LLaMA3b-8b. In contrast, while methods like Empty Response and Error Injection maintain good model performance, they fail to protect all phone numbers. Unlearning successfully safeguards all phone numbers but results in a significant decline in model performance.

Table 3 shows that our method, PPA applied to LLaMA2-7b for defense against physical address exposure, outperforms both Empty Response and Error Injection by reducing the risk score by 87.6% and 26.2%, respectively. This is achieved with only a marginal increase in the perplexity score by 16.7% and 35.4%, and a slight decrease in the Email Completion score by 30.7% and 29.4%. Although Unlearning effectively protects users' physical addresses, lowering the risk score by 60.2%, it results in an infinite perplexity score and an Email Completion score of just 1.0. Additionally, PPA

| Enron Email Experiment Phone Number Defense Model | | Model Performance | | Attack | | | | | | | |
|---|---|---|---|---|---|---|---|---|---|---|---|
| | | Perplexity | GPT-4o Email Score | Input Rephrase | | Probing | | Soft Prompt | | Attack Average | |
| | | | | RS ↓ | EM ↓ | RS ↓ | EM ↓ | RS ↓ | EM ↓ | RS ↓ | EM ↓ |
| LLaMA2-7b | Original | 5.6 | 8.6 | 1.3 | 1.0 | 0.0 | 0.0 | 0.0 | 0.0 | 0.4 | 0.3 |
| | Finetuned | 16.2 | 5.0 | 63.9 | 60.0 | 57.9 | 56.0 | 87.6 | 84.0 | 69.8 | 66.7 |
| | Empty Response | 16.5 | 5.7 | 51.7 | 49.3 | 37.2 | 34.8 | 80.5 | 75.9 | 56.4 | 53.3 |
| | Error Injection | 14.6 | 5.2 | 24.2 | 22.2 | 19.3 | 17.6 | 21.7 | 20.8 | 21.7 | 20.2 |
| | Unlearning | $3.2 \times 10^{11}$ | 1.1 | 0.0 | 0.0 | 0.0 | 0.0 | 0.0 | 0.0 | 0.0 | 0.0 |
| | DEPN | 77.2 | 2.0 | 9.0 | 7.7 | 0.0 | 0.0 | 8.2 | 6.4 | 5.7 | 4.7 |
| | PPA | 16.0 | 5.2 | 0.0 | 0.0 | 0.0 | 0.0 | 0.0 | 0.0 | 0.0 | 0.0 |
| LLaMA3-8b | Original | 9.9 | 9.2 | 0.0 | 0.0 | 0.0 | 0.0 | 0.0 | 0.0 | 0.0 | 0.0 |
| | Finetuned | 79.2 | 5.0 | 44.6 | 42.4 | 38.3 | 37.3 | 42.6 | 40.6 | 48.9 | 2.3 |
| | Empty Response | 82.9 | 4.7 | 25.7 | 23.8 | 21.5 | 19.8 | 25.3 | 24.1 | 24.1 | 22.5 |
| | Error Injection | 60.5 | 5.3 | 13.3 | 12.9 | 5.8 | 5.2 | 18.1 | 16.6 | 12.4 | 11.5 |
| | Unlearning | $5.0 \times 10^{21}$ | 1.2 | 0.0 | 0.0 | 0.0 | 0.0 | 0.0 | 0.0 | 0.0 | 0.0 |
| | DEPN | 138.5 | 4.5 | 37.2 | 34.2 | 26.1 | 24.5 | 26.7 | 25.3 | 30.0 | 28.0 |
| | PPA | 67.5 | 4.3 | 0.0 | 0.0 | 0.0 | 0.0 | 0.0 | 0.0 | 0.0 | 0.0 |

Table 2: Comparative Analysis of Phone Number Defense Strategies Against Various Attacks in Enron Email Experiment. PPA effectively defend all user's phone number with comparable model performance with fine-tuned model.

| Enron Email Experiment Physical Address Defense Model | | Model Performance | | Attack | | | | | | | |
|---|---|---|---|---|---|---|---|---|---|---|---|
| | | Perplexity | GPT-4o Email Score | Input Rephrase | | Probing | | Soft Prompt | | Attack Average | |
| | | | | RS ↓ | EM ↓ | RS ↓ | EM ↓ | RM ↓ | EM ↓ | RS ↓ | EM ↓ |
| LLaMA2-7b | Original | 5.6 | 8.6 | 24.5 | 3.0 | 17.1 | 3.0 | 6.5 | 1.0 | 16.0 | 2.3 |
| | Finetuned | 16.2 | 5.0 | 59.4 | 1.0 | 50.4 | 1.0 | 72.7 | 2.8 | 60.8 | 1.6 |
| | Empty Response | 16.7 | 5.2 | 57.5 | 1.0 | 46.3 | 1.0 | 73.7 | 3.7 | 59.2 | 1.9 |
| | Error Injection | 14.4 | 5.1 | 19.2 | 1.0 | 5.1 | 1.0 | 5.4 | 1.0 | 9.9 | 1.0 |
| | Unlearning | inf | 1.0 | 3.8 | 1.0 | 2.5 | 1.0 | 2.5 | 1.0 | 2.9 | 1.0 |
| | DEPN | 218.9 | 1.4 | 16.3 | 1.5 | 5.2 | 1.0 | 2.8 | 1.0 | 8.1 | 1.2 |
| | PPA | 19.5 | 3.6 | 12.1 | 1.0 | 4.7 | 1.0 | 5.2 | 1.0 | 7.3 | 1.0 |
| LLaMA3-8b | Original | 9.9 | 9.2 | 21.5 | 7.4 | 19.4 | 6.4 | 7.8 | 2.6 | 16.2 | 5.4 |
| | Finetuned | 79.2 | 5.0 | 66.1 | 4.6 | 41.2 | 1.0 | 39.5 | 1.5 | 48.9 | 2.3 |
| | Empty Response | 78.8 | 4.8 | 45.7 | 5.3 | 37.4 | 3.1 | 29.8 | 2.6 | 37.6 | 3.6 |
| | Error Injection | 52.5 | 4.2 | 25.3 | 1.0 | 10.0 | 1.0 | 18.2 | 2.0 | 17.8 | 1.3 |
| | Unlearning | inf | 1.0 | 3.1 | 1.0 | 2.2 | 1.0 | 2.2 | 1.0 | 2.5 | 1.0 |
| | DEPN | 201.5 | 1.5 | 45.5 | 3.0 | 15.7 | 2.0 | 9.1 | 5.3 | 17.8 | 3.4 |
| | PPA | 57.6 | 4.0 | 16.7 | 1.0 | 2.2 | 1.0 | 25.8 | 1.0 | 14.9 | 1.0 |

Table 3: Comparative Analysis of Physical Address Defense Strategies Against Various Attacks in Enron Email Experiment. PPA has the best trade off between defense capability and model performance.

outperforms DEPN by reducing the risk score by 9.8%, decreasing the perplexity score by 91.0%, and increasing the Email Completion score by 157.1%. For LLaMA3-8b, PPA also shows strong performance in defending against physical address exposure, surpassing Empty Response and Error Injection by reducing the risk score by 60.3% and 16.2%, respectively. It achieves this while slightly decreasing the perplexity score by 26.9% and increasing it by 9.7%, with only a marginal decrease in the Email Completion score by 16.6% and 4.7%. Although Unlearning remains effective, reducing the risk score by 83.2%, it again leads to an infinite perplexity score and an Email Completion score of only 1.0. PPA outperforms DEPN by reducing the risk score by 16.2%, lowering the perplexity score by 71.4%, and increasing the Email Completion score by 166.6%.

**Fraud email experiment.** Table 4 shows that PPA effectively protects the phone numbers of 50 persons, achieving a phone number risk score of 0.3 while maintaining model performance comparable to that of the Fine-tuned LLaMA2-7b model. Similarly, PPA safeguards the physical addresses of the other 50 persons, achieving an physical address risk score of 3.0 without compromising model performance relative to the Fine-tuned LLaMA2-7b model.

| Fraud Email Experiment Defense Model | | Model Performance | | Attack | | | | | | | |
|---|---|---|---|---|---|---|---|---|---|---|---|
| | | Perplexity | GPT-4o Email Score | Input Rephrase | | Probing | | Soft Prompt | | Attack Average | |
| | | | | RS↓ | EM↓ | RS↓ | EM↓ | RS↓ | EM↓ | RS↓ | EM↓ |
| LLaMA2-7b Phone Number Defense | Original | 4.06 | 1.0 | 0.0 | 0.0 | 0.0 | 0.0 | 0.0 | 0.0 | 0.0 | 0.0 |
| | Finetuned | 1.11 | 2.4 | 20.6 | 19.0 | 20.2 | 19.0 | 0.4 | 0.0 | 13.7 | 12.6 |
| | Empty Response | 1.11 | 2.2 | 15.6 | 13.0 | 13.6 | 12.0 | 0.4 | 0.0 | 9.8 | 8.3 |
| | Error Injection | 1.10 | 2.0 | 11.7 | 11.0 | 7.2 | 6.0 | 0.2 | 0.0 | 6.3 | 5.6 |
| | Unlearning | 1.33 | 1.6 | 0.0 | 0.0 | 0.0 | 0.0 | 0.0 | 0.0 | 0.0 | 0.0 |
| | DEPN | 1.18 | 2.8 | 5.4 | 4.0 | 6.6 | 5.0 | 0.0 | 0.0 | 4.0 | 3.0 |
| | PPA | 1.10 | 2.7 | 1.0 | 1.0 | 0.0 | 0.0 | 0.0 | 0.0 | 0.3 | 0.3 |
| LLaMA2-7b Physical Address Defense | Original | 4.06 | 1.0 | 4.7 | 0.0 | 2.4 | 0.2 | 1.4 | 0.0 | 2.8 | 0.0 |
| | Finetuned | 1.11 | 2.4 | 12.4 | 0.0 | 18.6 | 0.0 | 4.3 | 0.0 | 11.7 | 0.0 |
| | Empty Response | 1.11 | 2.3 | 13.2 | 0.0 | 9.3 | 0.0 | 4.7 | 0.2 | 9.0 | 0.0 |
| | Error Injection | 1.10 | 2.2 | 8.3 | 0.2 | 6.5 | 0.0 | 3.8 | 0.2 | 6.2 | 0.1 |
| | Unlearning | 27.54 | 1.0 | 1.3 | 0.0 | 1.2 | 0.0 | 1.2 | 0.0 | 1.2 | 0.0 |
| | DEPN | 1.94 | 1.3 | 3.1 | 0.0 | 1.6 | 0.0 | 0.7 | 0.0 | 1.8 | 0.0 |
| | PPA | 1.10 | 2.5 | 4.2 | 0.0 | 4.2 | 0.0 | 0.8 | 0.0 | 3.0 | 0.0 |

Table 4: Comparative Analysis of Phone Number and Physical Address Defense Strategies Against Various Attacks in Fraud Email Experiment.

| Phone Number Ablation Study | Model Performance | | | | Attack | | | | |
|---|---|---|---|---|---|---|---|---|---|
| | Enron perplexity first 512 | Enron perplexity stride 256 | GPT4 perplexity | Average | Score | Input Rephrase | Probing | Soft Prompt | Average |
| Proactive Privacy Amnesia | 26.7 | 6.8 | 14.4 | 16.0 | RS↓ | 0.0 | 0.0 | 0.0 | 0.0 |
| | | | | | EM↓ | 0.0 | 0.0 | 0.0 | 0.0 |
| Sensitivity Analysis + Selective Forgetting | 853.6 | 11.8 | 28.0 | 297.8 | RS↓ | 0.0 | 0.0 | 0.0 | 0.0 |
| | | | | | EM↓ | 0.0 | 0.0 | 0.0 | 0.0 |
| Unlearning + Memory Implanting | 26.8 | 6.7 | 15.3 | 16.3 | RS↓ | 0.5 | 0.0 | 0.0 | 0.2 |
| | | | | | EM↓ | 0.5 | 0.0 | 0.0 | 0.2 |
| Fix index 0 Selective Privacy Amnesia | 24.7 | 6.5 | 14.4 | 15.2 | RS↓ | 28.8 | 23.6 | 27.8 | 26.7 |
| | | | | | EM↓ | 26.3 | 21.3 | 26.3 | 24.6 |
| Fix index 1 Selective Privacy Amnesia | 26.4 | 6.7 | 14.4 | 15.9 | RS↓ | 3.5 | 1.2 | 2.2 | 2.3 |
| | | | | | EM↓ | 3.0 | 1.0 | 2.0 | 2.0 |
| Fix index 2 Selective Privacy Amnesia | 24.9 | 6.6 | 14.6 | 15.4 | RS↓ | 0.4 | 1.0 | 1.8 | 1.1 |
| | | | | | EM↓ | 0.0 | 1.0 | 1.3 | 0.8 |

Table 5: Ablation study on phone numbers.

**Main Results.** We summarize the key results in Tables 2, 3, 4, as follows: Observations from both phone number and physical address defenses indicate that PPA provides the best balance between safeguarding users' PII and maintaining model performance, compared to other defense methods.

# 6 ABLATION STUDIES

## 6.1 ANALYSIS OF THREE STAGES IN PPA

**Sensitivity Analysis + Selective Forgetting.** To assess the effectiveness of sensitivity analysis combined with selective forgetting in preserving the PII of targeted persons, we applied this approach to protect PII, such as phone numbers and physical addresses. While selective forgetting marginally reduces performance degradation, the resulting model remains largely ineffective, as demonstrated in Table 5 and Table 6.

| Physical Address Ablation Study | Model Performance | | | | Attack | | | | |
|---|---|---|---|---|---|---|---|---|---|
| | Enron perplexity first 512 | Enron perplexity stride 256 | GPT4 perplexity | Average | Score | Input Rephrase | Probing | Soft Prompt | Average |
| Proactive Privacy Amnesia | 35.6 | 8.2 | 14.6 | 19.5 | RS↓ | 12.1 | 4.7 | 5.2 | 7.3 |
| | | | | | EM↓ | 1.0 | 1.0 | 1.0 | 1.0 |
| Sensitivity Analysis + Selective Forgetting | inf | $8.2 \times 10^{30}$ | $1.6 \times 10^{18}$ | inf | RS↓ | 3.8 | 2.5 | 2.5 | 2.9 |
| | | | | | EM↓ | 1.0 | 1.0 | 1.0 | 1.0 |
| Unlearning + Memory Implanting | 61.4 | 10.6 | 28.8 | 33.6 | RS↓ | 10.1 | 4.9 | 5.2 | 6.7 |
| | | | | | EM↓ | 1.0 | 1.0 | 1.0 | 1.0 |
| Fix index 0 Selective Privacy Amnesia | 29.9 | 7.4 | 20.6 | 19.3 | RS↓ | 34.3 | 9.8 | 7.5 | 17.2 |
| | | | | | EM↓ | 1.0 | 1.0 | 1.0 | 1.0 |
| Fix index 1 Selective Privacy Amnesia | 29.8 | 7.6 | 15 | 17.4 | RS↓ | 11.7 | 5.5 | 15.4 | 10.9 |
| | | | | | EM↓ | 1.0 | 1.0 | 1.0 | 1.0 |
| Fix index 2 Selective Privacy Amnesia | 53.3 | 9.4 | 14.9 | 25.9 | RS↓ | 5.7 | 3.1 | 4.5 | 4.4 |
| | | | | | EM↓ | 1.0 | 1.0 | 1.0 | 1.0 |

Table 6: Ablation study on physical addresses.

**Fix index Selective Privacy Amnesia.** To evaluate the efficacy of sensitivity analysis in safeguarding the PII of targeted persons, we employed a fixed index Selective Privacy Amnesia approach, focusing on indices 0, 1, and 2 as the primary elements for removal. Our findings indicate that the fixed index Selective Privacy Amnesia falls short in adequately safeguarding users' phone numbers, as illustrated in Table 5. When it comes to preserving users' physical addresses, as depicted in Table 6, employing the fixed index 0 and 1 Selective Privacy Amnesia, yielding address risk scores of 17.2 and 10.9 respectively, does not offer as robust protection as Proactive Privacy Amnesia, which yields an address risk score of 7.3. While the fixed index 2 Selective Privacy Amnesia, with an address risk score of 4.4, does provide superior protection compared to Proactive Privacy Amnesia, it comes with higher perplexity, indicative of lower model performance. This is attributed to the fixed index potentially altering the original meaning of words. For instance, in the case of "New York City," unlearning the single token "City" would compel the model to disregard the frequent occurrence of "City" following "New York," consequently compromising the model's performance.

**Unlearning + Memory Implanting.** To assess the effectiveness of sensitivity analysis coupled with selective forgetting in safeguarding the PII of targeted persons, we implemented unlearning in conjunction with memory implanting, as shown in Table 5 6. Our findings revealed that Unlearning + Memory Implanting proved capable of safeguarding the majority of persons' phone numbers and physical addresses, resulting in phone number risk scores of 0.2 and address risk scores of 6.7. However, this approach exhibited higher perplexity levels, measuring 16.3 and 33.6, which signifies diminished model performance. This is because the unlearning method essentially erases entire PII sequences, thereby enhancing PII protection capabilities at the expense of model performance.

## 6.2 PPA Trade-Offs: Number of Forgotten Indexes

Table 2 and 3 reveal that, after applying PPA, the address risk score remains at 7.3, while the phone risk score drops to 0. This disparity may be due to physical addresses being longer and less structured than phone numbers. Therefore, we have conducted an ablation study to quantify the drop-off in risk score as the number of forgotten indexes increases. We conducted experiments on Address PPA. Specifically, we set $k = 1, 5, 10, 15, 20, 25$ in Equations 5 and 6. Both stages followed the training protocols in Appendix U for a single epoch. For Addresses, the PPA method improves the PII risk score if more than one index is selected for forgetting. However, selecting too many indexes causes the model performance to deteriorate, as shown in Figure 3. This ablation demonstrates that PPA is a flexible method, allowing for adjustments to the balance between defense capability and model performance by modifying the number of key elements to be forgotten.

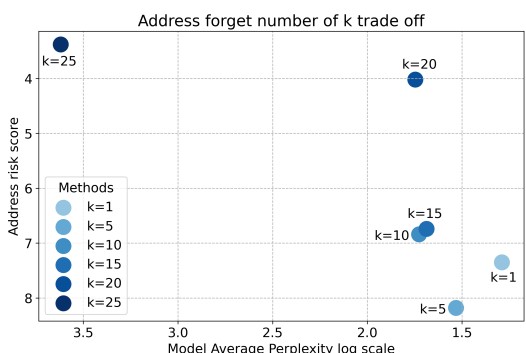

Figure 3: Address PPA Risk score vs forget number of indexes: PPA tunes the parameter $k$, as defined in Equations 5 and 6.

## 7 Conclusion and Discussion

We demonstrated that Proactive Privacy Amnesia achieves the optimal balance between defense performance and model utility compared to methods like Error Injection, Empty Response, Unlearning, and DEPN for protecting users' PII, including phone numbers and physical addresses. Additionally, we initially introduce the concept of the 'memorization factor', which affects the model's capacity to retain PII sequences. This concept is using in sensitivity analysis and supported by theoretical justification. Furthurmore, PPA is a flexible method that can adjust its balance between defense capability and model performance. Future work could extend PPA to protect the privacy of relationships, such as those between persons or between organizations.

# 8 ACKNOWLEDGEMENTS

This work is supported by ARO W911NF-23-2-0224, NSF IIS-2332744, and CNS-2112562. We also thank Accenture for its support and Yun-Chu Wang for her support. We thank area chair and reviewers for their valuable comments.

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

APPENDIX

## A  EXISTING ASSETS

The two existing assets used in this work are the Enron email dataset Klimt & Yang (2004) and aeslc dataset Zhang & Tetreault (2019). Please see their information below.

- Enron email dataset: The URL is http://www.enron-mail.com/email/; the license is not clearly stated by the authors.
- aeslc dataset: The URL is https://huggingface.co/datasets/aeslc; the license is not clearly stated by the authors.

## B  DETAILS OF BUILDING GROUND TRUTH TABLE

To evaluate our defense method, we constructed a ground truth table comprising two parts: (1) We utilized the AWS Comprehend Service (Amazon Web Services Comprehend, 2024) to extract PII, including names, phone numbers, and physical addresses; (2) we employed (Manakul et al., 2023) to determine the correlations between specific PIIs and the corresponding persons.

## C  DETAILS OF EVALUATION DATASET

For Enron email dataset, we constructed our evaluation ground truth table using the aelsc training dataset (Zhang & Tetreault, 2019). However, there is overlap between the aelsc training and validation datasets, which are used to train the soft prompt for the soft prompt attack. To ensure a fair comparison between soft prompt and probing attacks, we excluded certain persons' scores, as detailed in Appendix D. Consequently, our evaluation focused on 468 persons whose phone numbers were disclosed and 790 persons whose physical addresses were revealed.

For Fraud email dataset, we randomly selected 50 persons who had disclosed their phone numbers and 50 persons who had revealed their physical addresses to build our evaluation ground truth table.

## D  EVALUATION DETAILS OF THE ENRON EMAIL EXPERIMENT.

We constructed our evaluation ground truth table on the aeslc training dataset (Zhang & Tetreault, 2019), which comprises data from 1,359 persons. Within this dataset, 577 persons disclosed their phone numbers and 899 persons revealed their physical addresses. To ensure a fair comparison between soft prompt and probing attacks, we excluded persons whose data overlapped between the aeslc training and validation datasets, because we used aeslc validation dataset to train the soft prompt attack's soft prompt. The number of overlapping persons is 109. Consequently, our evaluation focused on 468 persons (577 - 109) whose phone numbers were exposed and 790 persons (899 - 109) whose physical addresses were exposed. Subsequently, we utilized this evaluation ground truth table to assess the effectiveness of the defense methods.

## E  COMPARISON WITH DIFFERENTIAL PRIVACY-BASED METHODS

We implemented the Differentially Private Decoding (DP Decoding) in (Majmudar et al., 2022) and the Just Fine-Tune Twice (JFT) method in (Shi et al., 2022). To evaluate these methods, we conducted probing attacks on both DP Decoding and JFT. Specifically, for DP Decoding, the parameter $\lambda$ impacts both utility (the effectiveness of the generated output) and privacy (the protection of sensitive information). As $\lambda$ varies from 0 to 1, a value closer to 0 enhances privacy but reduces utility, while a value closer to 1 improves utility at the expense of privacy. We tested various values of the $\lambda$ parameter ranging from (0.1, 0.3, 0.5, 0.7, 0.9) and we selected $\lambda$=0.3 as it achieved the best balance between utility and privacy protection among the tested values, offering better utility compared to $\lambda$=0.1, and improved defense capability compared to $\lambda$=0.5, as shown in Table 7. We observed that our PPA method still outperformed both DP Decoding and JFT, achieving a lower risk

score and a higher utility score, as shown in Table 8. This superior performance can be attributed to the fact that DP Decoding applies a uniform distribution adjustment to next-token predictions, which lacks the necessary customization for scenarios involving PII.

| | PPA | $\lambda$=0.1 | $\lambda$=0.3 | $\lambda$=0.5 | $\lambda$=0.7 | $\lambda$=0.9 | no defense |
|---|---|---|---|---|---|---|---|
| Risk Score ↓ | 0.0 | 15.7 | 30.4 | 32.4 | 33.2 | 41.4 | 57.9 |
| Exact Match Score ↓ | 0.0 | 10.6 | 28.4 | 30.1 | 31.1 | 38.6 | 56.0 |
| GPT-4o Email Score | 5.2 | 3.0 (3.025) | 4.8 (4.750) | 4.8 (4.825) | 4.9 (4.875) | 5.1 (5.125) | 5.0 |

Table 7: Comparison of Different $\lambda$ Values in DP Decoding. $\lambda$=0.3 achieved the best balance between utility and privacy protection among the tested $\lambda$ value.

| Phone Defense Model | Risk Score ↓ | Exact Match Score ↓ | GPT-4o Email Score |
|---|---|---|---|
| DP Decoding | 30.4 | 28.4 | 4.8 |
| JFT | 28.4 | 26.0 | 5.0 |
| PPA | 0.0 | 0.0 | 5.2 |

Table 8: Performance comparison of Differential Privacy-Based methods.

# F    MODEL PERFORMANCE EVALUATION ON MMLU AND TRUTHFULQA

We provide some new evaluations on the model's performance metrics on both MMLU (Hendrycks et al., 2020) and TruthfulQA (Lin et al., 2021). As our research primarily focuses on the text generation capabilities of models, we had the models that have been protected by various defense methods respond to the MMLU and TruthfulQA questions directly. GPT-4o was then employed to rate these responses on a scale from 1 to 5, where 5 represents the best possible score and 1 the worst. Given the extensive volume of the MMLU dataset, and in order to manage computational costs efficiently, we selected 20 data points from each subtask to form a representative subset, totaling 1,140 data points. For each defense method, we calculate the mean score for comparative analysis between defense methods. We found that PPA achieves the highest MMLU and TruthfulQA score among all baseline defense methods, as illustrated in Table 9.

| Phone Defense Model | MMLU Score | TruthfulQA Score | GPT-4o Email Score |
|---|---|---|---|
| Empty Response | 3.3 | 3.4 | 5.7 |
| Error Injection | 3.3 | 3.2 | 5.2 |
| Unlearning | 1.6 | 1.7 | 1.1 |
| DEPN | 2.3 | 2.4 | 2.0 |
| PPA | 4.2 | 4.1 | 5.2 |

Table 9: Comparison of Phone Defense Models based on MMLU and TruthfulQA mean scores and GPT-4o Email Scores.

# G    STRONGER ATTACKER HAS PRIOR KNOWLEDGE OF THE PII

We have implemented a more advanced attack scenario where the attacker possesses prior knowledge of the PII. Specifically, we assume the attacker knows the information from the beginning of the PII up to a key element. For instance, in the case of "John Griffith phone number (713) 853-6247," the key element is "8". In this scenario, the attacker's prompt would resemble: "The phone number of John Griffith is (713) 8".

As shown in the Table 10, we observe that PPA achieves the best balance between defense capability and model performance.

| Phone Defense Model | Risk Score ↓ | Exact Match Score ↓ | GPT-4o Email Score |
|---|---|---|---|
| Empty Response | 154.0 | 141.5 | 5.7 |
| Error Injection | 75.1 | 69.1 | 5.2 |
| Unlearning | 6.7 | 1.8 | 1.1 |
| DEPN | 36.2 | 27.3 | 2.0 |
| PPA | 12.1 | 9.8 | 5.2 |

Table 10: Phone Defense Models against strong attacker has the prior knowledge.

## H  DETAILS OF THE PROPORTIONS OF KEY ELEMENTS

We calculated the proportions of key element lengths relative to the total lengths for phone numbers and physical addresses, which are 6.7% and 27.6%, respectively.

## I  MORE DISCUSSION ABOUT MEMORY IMPLANTING

We modified the memory implanting component to focus on replacing the key element with a different token. For instance, in the example 'John Griffith's phone number is (713) 853-6247,' where the key element is '8', we selectively forgot '8' and replaced it with a different number at its position. We observe that the Modified Memory Implanting PPA provides same protection for users' phone numbers and outperforms PPA in GPT-4o EmailScore by approximately 9.6%, as shown in Table 11. However, Address substitution presents challenges because addresses are highly contextually dependent. Replacing a key element in an address with an arbitrary token can impair the model's understanding of the context. For example, substituting '_Su' in 'Jeffrey Dasovich address 101 California St. Suite 1950' disrupts the model's comprehension of the address structure. Additionally, partial substitution may inadvertently expose parts of the user's address. Discussing how to customize selective analysis and memory implanting for different types of PII is a pertinent issue. Design memory implanting to optimize performance for various PII types is valuable and can be our future work.

| Phone Defense Model | Risk Score ↓ | Exact Match Score ↓ | GPT-4o Email Score |
|---|---|---|---|
| Empty Response | 37.2 | 34.8 | 5.7 |
| Error Injection | 19.3 | 17.6 | 5.2 |
| Unlearning | 0.0 | 0.0 | 1.1 |
| DEPN | 0.0 | 0.0 | 2.0 |
| PPA | 0.0 | 0.0 | 5.2 |
| Modified Memory Implanting PPA | 0.0 | 0.0 | 5.7 |

Table 11: Comparison of the Modified Memory Implanting PPA with Other Phone Defense Strategies.

## J  ADDING THE EXPOSURE METRIC

We calculated the exposure metric (Carlini et al., 2019) for all baseline methods. Since calculating the exposure of PII is computationally intensive, we followed the approach in Table 2 of (Carlini et al., 2019) and evaluated the exposure for 10 phone numbers. Our results show that PPA outperforms other baseline defense methods, as shown in Table 12

| Phone Defense Model | Exposure |
|---|---|
| Empty Response | 12.50 |
| Error Injection | 10.94 |
| Unlearning | 3.55 |
| DEPN | 7.72 |
| PPA | 0.05 |

Table 12: Exposure levels of various Phone Defense Strategies.

## K  EVALUATION ON EMAIL ADDRESS.

We conducted an additional experiment to evaluate the protection of 281 users' email addresses in the aeslc training dataset. Using Levenshtein distance (Po, 2020), we compared the predicted email addresses to the ground truth. As shown in the Table 13, PPA successfully defends all users' email addresses against probing attacks while maintaining model performance comparable to other baseline defense methods.

| Email Defense Model | Risk Score ↓ | Exact Match Score ↓ | GPT-4o Email Score |
|---|---|---|---|
| Empty Response | 47.2 | 40.5 | 5.1 |
| Error Injection | 19.6 | 17.0 | 5.3 |
| Unlearning | 1.0 | 1.0 | 1.6 |
| DEPN | 1.0 | 1.0 | 1.3 |
| PPA | 0.0 | 0.0 | 5.0 |

Table 13: Comparative Analysis of Email Defense Strategies Against Various Attacks in Enron Email Experiment.

## L  ORIGINALLY SAFE INFORMATION TO BE EXPOSED?

Motivated by the concern that the PPA defense could inadvertently expose previously secure PII of users who are not explicitly protected by the method. We evaluated the exposure metric (Carlini et al., 2019) for safe phone numbers—those not exposed to attackers—that were not protected by the PPA method, using both the no-defense setup and the PPA model. Given the time-intensive nature of calculating PII exposure, we referenced Table 2 from The Secret Sharer (Carlini et al., 2019) and analyzed the exposure of 10 such phone numbers. The average exposure for these cases is summarized in the Table 14.

As shown in the Table 14, the exposure of phone numbers not protected by the PPA method decreases slightly, from 1.57 (no defense) to 1.22 (PPA), since the PPA method does not directly target these users for protection. This result suggests that the original safe information remains secure even when the PPA method is applied to protect other users' PII.

| Phone Defense Model | Exposure |
|---|---|
| No Defense | 1.57 |
| PPA | 1.22 |

Table 14: Exposure of Users' Phone Numbers not protected by the PPA Method.

## M    DISCUSS SCALABILITY FOR PPA

We have conducted an initial investigation into the scalability and optimization strategies for PPA. Our experiments involved combining PPA with efficient fine-tuning techniques, such as LoRA (Hu et al., 2021), using a rank of 16 and an alpha value of 32. As shown in the Table 15, applying LoRA to PPA produced promising results: after fine-tuning for three epochs, the risk score reduced to 1.0, and after four epochs, it further decreased to 0.0, all while maintaining comparable model performance. Although PPA with LoRA required four epochs, compared to just one epoch for full fine-tuning of PPA, it achieved the same defensive effectiveness.

Table 15 demonstrates that PPA has potential for scalability. Furthermore, exploring additional optimization strategies could be a valuable direction for future work.

| Phone Defense Model | Risk Score ↓ | Exact Match Score ↓ | GPT-4o Email Score |
|---|---|---|---|
| PPA LoRA 1-epoch | 24.6 | 23.6 | 5.4 |
| PPA LoRA 2-epoch | 5.4 | 5.2 | 5.2 |
| PPA LoRA 3-epoch | 1.0 | 1.0 | 5.0 |
| PPA LoRA 4-epoch | 0.0 | 0.0 | 5.1 |

Table 15: Comparative Analysis of PPA LoRA with different fine-tuning epochs.

## N    ABLATION STUDY ON VARIOUS FORGETTING STRATEGIES

We conducted ablation studies on protecting the user's physical address using the following three strategies: 1) Only the most sensitive tokens (PPA), 2) Span around the most sensitive token, and 3) The most sensitive tokens + the following tokens. Their respective Risk Scores and GPT-4o Email Scores are presented in Table 16.

As shown in the Table 16, "Only the most sensitive token (PPA)" achieves an optimal balance between defense capability and model performance. "Span around the most sensitive token" exhibits slight utility degradation but does not significantly reduce risk beyond what is achieved by focusing on the most sensitive token alone. "The most sensitive tokens + the following tokens" strategy reduces the risk score further but at the cost of significantly lowering the GPT-4o Email Score. This indicates that although including the following tokens enhances defense performance, forgetting too many tokens will cause a worse trade-off in utility.

Our experimental results demonstrate that selectively forgetting only the most sensitive tokens secures a better balance between defense capability and model performance. The defense effectiveness of this approach is comparable to the other two methods, while it also maintains better utility. Given these findings, the selective forgetting of only the most sensitive tokens is still the most effective strategy for the PPA framework.

| Forgetting Strategies | Risk Score ↓ | GPT-4o Email Score |
|---|---|---|
| Only the most sensitive tokens (PPA) | 4.7 (4.702) | 3.6 |
| Span around the most sensitive token | 4.7 (4.737) | 3.4 |
| The most sensitive tokens + the following tokens | 4.4 | 2.9 |

Table 16: Comparative Analysis of various forgetting strategies.

| Enron Email Experiment | | Model Performance | | | |
|---|---|---|---|---|---|
| **Phone Number Defense Model** | | Enron perplexity first=512 tokens | Enron perplexity stride=256 | GPT4 perplexity | Perplexity |
| LLaMA2-7b | Original | 7.6 | 3.2 | 6.2 | 5.6 |
| | Finetuned | 26.1 | 6.6 | 16.0 | 16.2 |
| | Empty Response | 26.7 | 6.7 | 16.2 | 16.5 |
| | Error Injection | 23.2 | 6.3 | 14.2 | 14.6 |
| | Unlearning | $9.7 \times 10^{11}$ | 5691 | $1.9 \times 10^6$ | $3.2 \times 10^{11}$ |
| | DEPN | 139.5 | 19.9 | 72.3 | 77.2 |
| | PPA | 26.7 | 6.8 | 14.4 | 16.0 |
| LLaMA3-8b | Original | 10.5 | 4.4 | 15.0 | 9.9 |
| | Finetuned | 58.9 | 11.4 | 167.4 | 79.2 |
| | Empty Response | 61.7 | 11.7 | 175.5 | 82.9 |
| | Error Injection | 46.5 | 9.9 | 125.2 | 60.5 |
| | Unlearning | $1.5 \times 10^{22}$ | $8.5 \times 10^{14}$ | $1.1 \times 10^{14}$ | $5.0 \times 10^{21}$ |
| | DEPN | 92.2 | 14.9 | 308.6 | 138.5 |
| | PPA | 57.5 | 11.3 | 133.8 | 67.5 |

Table 17: Comparative Analysis of Perplexity Against Various Attacks in Enron Email Experiment. PPA stands for Proactive Privacy Amnesia. 'Perplexity' refers to the average of our perplexity metric.

| Enron Email Experiment | | Model Performance | | | |
|---|---|---|---|---|---|
| **Physical Address Defense Model** | | Enron perplexity first=512 tokens | Enron perplexity stride=256 | GPT4 perplexity | Perplexity |
| LLaMA2-7b | Original | 7.6 | 3.2 | 6.2 | 5.6 |
| | Finetuned | 26.1 | 6.6 | 16.0 | 16.2 |
| | Empty Response | 27.3 | 6.7 | 16.1 | 16.7 |
| | Error Injection | 23.5 | 6.4 | 13.3 | 14.4 |
| | Unlearning | inf | $3.1 \times 10^{31}$ | $4.73 \times 10^{21}$ | inf |
| | DEPN | 480.4 | 43.8 | 132.6 | 218.9 |
| | PPA | 35.6 | 8.2 | 14.6 | 19.5 |
| LLaMA3-8b | Original | 10.5 | 4.4 | 15.0 | 9.9 |
| | Finetuned | 58.9 | 11.4 | 167.4 | 79.2 |
| | Empty Response | 60.0 | 11.5 | 165.1 | 78.8 |
| | Error Injection | 44.8 | 9.7 | 103.2 | 52.5 |
| | Unlearning | $1.2 \times 10^{27}$ | $1.3 \times 10^{18}$ | inf | inf |
| | DEPN | 177.8 | 23.2 | 403.7 | 201.5 |
| | PPA | 63.0 | 12.4 | 97.4 | 57.6 |

Table 18: Comparative Analysis of Perplexity Against Various Attacks in Enron Email Experiment. PPA stands for Proactive Privacy Amnesia. 'Perplexity' refers to the average of our perplexity metric.

## O  DETAILS OF MODEL PERFORMANCE PERPLEXITY METRIC

In Tables 2, 3, and 4, we report the average Perplexity across three different tests to evaluate model performance. The detailed results of each individual Perplexity test are provided in Tables 17, 18, and 19.

## P  DETAILS OF THE BASELINE DEFENSE METHODS

**Empty Response** (Patil et al., 2023; Ouyang et al., 2022)**.** This method refines the model to label non-sensitive information as "dummy". For instance, we create templates for each person, formatted with the person's name, PII type, and "dummy". We then perform gradient descent on "dummy" following the training settings outlined in Appendix U with a single epoch.

**Error Injection.** We implemented the Error Injection method on each person's phone numbers, conducting a single epoch of training. This same process is used to preserve a person's physical addresses. Take a person's phone number as an example, we create templates for each person,

| Fraud Email Experiment | | Model Performance | | |
|---|---|---|---|---|
| **Defense Model** | | Enron perplexity first=512 tokens | Enron perplexity stride=256 | Perplexity |
| LLaMA2-7b Phone Number Defense | Original | 5.33 | 2.79 | 4.06 |
| | Finetuned | 1.17 | 1.05 | 1.11 |
| | Empty Response | 1.17 | 1.05 | 1.11 |
| | Error Injection | 1.15 | 1.05 | 1.10 |
| | Unlearning | 1.53 | 1.14 | 1.33 |
| | DEPN | 1.27 | 1.09 | 1.18 |
| | PPA | 1.16 | 1.05 | 1.10 |
| LLaMA2-7b Physical Address Defense | Original | 5.33 | 2.79 | 4.06 |
| | Finetuned | 1.17 | 1.05 | 1.11 |
| | Empty Response | 1.17 | 1.05 | 1.11 |
| | Error Injection | 1.15 | 1.05 | 1.10 |
| | Unlearning | 48.21 | 1.14 | 1.33 |
| | DEPN | 2.48 | 1.09 | 1.18 |
| | PPA | 1.16 | 1.05 | 1.10 |

Table 19: Comparative Analysis of Perplexity Against Various Attacks in Enron Email Experiment. PPA stands for Proactive Privacy Amnesia. 'Perplexity' refers to the average of our perplexity metric.

structured as the person's name, PII type, and fake PII, which is generated by (Microsoft, 2024). We then apply gradient descent to false PII, adhering to the training settings detailed in Appendix U.

**Unlearning** (Jang et al., 2022). We applied an unlearning technique to the PII sequence by performing gradient ascent on it, following the training settings specified in Appendix U with a single epoch.

**DEPN** (Wu et al., 2023) We adopted the DEPN approach, as detailed in the DEPN GitHub repository, to protect PII, specifically phone numbers and physical addresses. Our goal was to eliminate specific neurons from the output of the LlamaDecoderLayer in the LlamaModel (Meta-Llama). We established a threshold ratio of 0.01 for both phone numbers and physical addresses, with mode ratio bags set at 0.49 and 0.5, respectively. Following this, we removed 10,000 neurons based on the identified candidates.

## Q    DETAILS OF ATTACK METHODS

For the input rephrasing attack, we generated 20 attack templates based on the twin template described in (Kim et al., 2024).

For soft prompt tuning in the Enron email experiment, we used the first probing twin template from (Kim et al., 2024), leveraging the aeslc validation ground truth table. In the Fraud email experiment, we applied the same probing template, selecting 25 persons with phone numbers and 25 with physical addresses randomly from the fraud email dataset.

## R    DETAILS OF ATTACK SUCCESS METRIC

For the total phone numbers and physical address risk score, we calculate the average phone number risk score and average physical address risk score for each person. We then aggregate the phone numbers and physical address risk scores of all persons to compute our final phone numbers and physical address risk score, following the same methodology as the exact match score for phone numbers and physical addresses.

## S    MODEL PERFORMANCE METRIC

For the **Perplexity metric**, we conducted three different tests to assess model performance. First, we calculated perplexity (Face, 2024) on the first 512 tokens of each text (with a maximum length of 512 tokens). Second, we computed the perplexity for each letter, using a maximum length of 512 tokens

and a stride of 256 tokens. Third, we assessed the perplexity of letters generated by GPT-4 (Achiam et al., 2023), but the Fraud email experiment did not have this metric, because GPT-4 cannot write fraud emails due to its safety aligned mechanism. These three tests help us determine whether our defense method impacts model performance.

For the **Email completion metric** in the Enron email experiment, we evaluated the model's performance in completing truncated emails. Specifically, we tasked the model with completing 40 truncated emails, which were subsequently evaluated by GPT-4o (OpenAI, 2024). Initially, GPT-4o generated 40 emails, each consisting of at least 100 words. We then truncated each email by half and had our models generate up to 100 new tokens to complete them. GPT-4o assessed and ranked the completions on a scale of 1 to 10, with 10 representing the best score. The average score was calculated across all 40 completions.

For the **Email completion metric** in the Fraud email experiment, we evaluated the model's ability to generate complete fraud emails. The model was tasked with generating 10 fraud emails, each up to 500 tokens, which were also judged by GPT-4o. GPT-4o ranked these completions on the same 1 to 10 scale, with the average score being calculated across all 10 fraud email completions.

# T  PROOF

## T.1  PROOF OF PROPOSITION 1

**Proposition 1.** *Maximizing the memorization factor can lead to*

$$\max_k D(k) = \begin{cases} \max_k L(k) & \text{if } \exists k, \nabla L(k) = 0, \\ \max_k 1/d_{Newton}(k) & \text{if } \nexists k, \nabla L(k) = 0. \end{cases} \tag{8}$$

$d_{Newton}(k)$ *is Newton's Direction at k, which is from Newton Method in convex optimization (Boyd & Vandenberghe, 2004).* $\max_k 1/d_{Newton}(k)$ *is achieved when* $d_{Newton}(k) \to 0^+$*. As L(k) is monotonically non-decreasing, a small positive* $d_{Newton}(k)$ *implies that the gradient at token k quickly decreases with a negative second-order derivative.*

*Proof.* Notice that

$$L(k) = - \sum_{\boldsymbol{x}_1 \cdots, \boldsymbol{x}_k} p(\boldsymbol{x}_1 \cdots, \boldsymbol{x}_k) \log q(\boldsymbol{x}_1 \cdots, \boldsymbol{x}_k) \tag{9}$$

$$= - \sum_{\boldsymbol{x}_1 \cdots, \boldsymbol{x}_{k-1}} p(\boldsymbol{x}_1 \cdots, \boldsymbol{x}_{k-1}) \log q(\boldsymbol{x}_1 \cdots, \boldsymbol{x}_{k-1})$$

$$- \sum_{\boldsymbol{x}_1 \cdots, \boldsymbol{x}_{k-1}} p(\boldsymbol{x}_1 \cdots, \boldsymbol{x}_{k-1}) \sum_{\boldsymbol{x}_k} p(\boldsymbol{x}_k | \boldsymbol{x}_1 \cdots, \boldsymbol{x}_{k-1}) \log q(\boldsymbol{x}_k | \boldsymbol{x}_1 \cdots, \boldsymbol{x}_{k-1}) \tag{10}$$

$$= L(k-1) + H_k, \tag{11}$$

So we have

$$H_k = L(k) - L(k-1) \approx \nabla L(k), \tag{12}$$

$$H_{k+1} - H(k) \approx \nabla L(k+1) - \nabla L(k) \approx \nabla^2 L(k), \tag{13}$$

$$D_k = \frac{H_k - H_{k+1}}{H_k} \approx -\frac{\nabla^2 L(k)}{\nabla L(k)}. \tag{14}$$

Our selection method selects $k$ with the largest $D_k$. We discuss it in two situations:

1. When there exists $k$ such that $H_k = \nabla L(k) = 0$, we require that $\nabla^2 L(k) < 0$ to achieve the maximum ($D_k = +\infty$), this guarantees that $k$ achieves the maximum of $L(k)$ as well.

2. When $H_k$ is always positive (notice that $H_k$ is never negative), $L(k)$ keeps growing as $k$ increases so we cannot find the maximum. But we still have

$$\max_k D_k = \max_k \frac{1}{d_{\text{Newton}}(k)} = \min_k d_{\text{Newton}}(k), \tag{15}$$

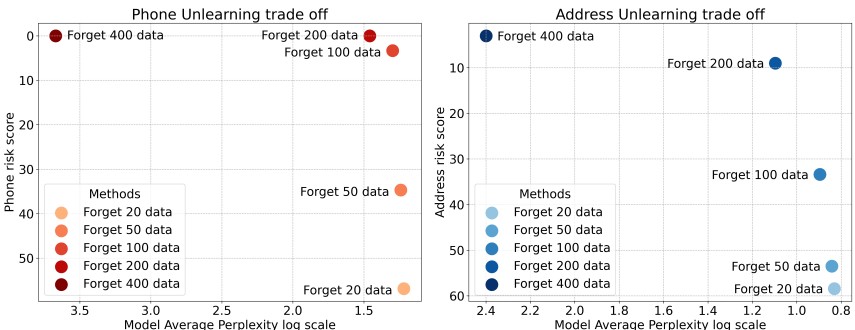

Figure 4: Unlearning method trade-off: Risk score vs forget number of data. left: phone numbers; right: physical addresses

where $d_{\text{Newton}}(k) = -\nabla L(k)/\nabla^2 L(k)$ is *Newton's Direction* in the second-order Newton's Method. The maximization is achieved when $d_{\text{Newton}}(k) \to 0^+$. Since $\nabla L(k) > 0$, $d_{\text{Newton}}(k) \to 0^+$ is achieved when $\nabla^2 L(k) = -\infty$, which implies that the gradient at $k$ quickly approaches $0$.

$\square$

## U  TRAINING SETTING AND HARDWARE

The training settings for fine-tuning the LLM on the Enron and Fraud email datasets, as well as for implementing defensive methods such as gradient descent and ascent, are as follows: a batch size of 4, the AdamW optimizer, a learning rate of 5e-5, weight decay of 0.001, a cosine learning rate scheduler, and a warmup ratio of 0.03. All experiments were conducted using 8 NVIDIA Quadro RTX 6000 24GB GPUs.

## V  ADDITIONAL ANALYSIS ON UNLEARNING SCALING EXPERIMENT

### V.1  UNLEARNING METHOD TRADE-OFF

To analyze the break-even point of the unlearning method, we conducted experiments focusing on both phone numbers and address unlearning. We tested the forgetting of 20, 50, 100, 200, and 400 data points. The results indicate that as more data points are forgotten, a greater number of phone numbers and physical addresses are preserved. However, this leads to a deterioration in the model's performance, as illustrated in Figure 4. We discovered that forgetting between 200 and 400 data points significantly increases perplexity and it indicated that the break-even point for the unlearning method is when between 200 and 400 data points are forgotten.

## W  BROADER IMPACTS.

The societal implications of our work include positive impacts, as it can protect PII from fine-tuned LLMs with only a negligible drop in performance, ensuring that the LLMs remain effective for their intended purposes. And it is unlikely to cause significant negative societal impacts.

