# OpenReview forum: "Proactive Privacy Amnesia for Large Language Models: Safeguarding PII with Negligible Impact on Model Utility"
_ICLR.cc/2025/Conference — ICLR 2025 Poster_

### Official Review · Reviewer_J1tm · 2024-11-02

**Soundness:** 3
**Presentation:** 2
**Contribution:** 3
**Rating:** 6
**Confidence:** 3

**Summary:**

The article introduces a novel method to protect personally identifiable information in language models without compromising performance. Inspired by amnesia research, the PPA approach selectively forgets sensitive PII while preserving model utility. It involves three main steps: sensitivity analysis to identify critical elements within PII, selective forgetting of these elements, and memory implanting with non-sensitive information to maintain functionality. Experimental results on phone numbers and addresses show that PPA effectively reduces privacy risks and maintains high model performance compared to other methods.

**Strengths:**

-   The approach is distinctive as it draws from cognitive science concepts, such as anterograde amnesia, to design a targeted forgetting mechanism. This use of selective forgetting combined with memory implanting to balance utility and privacy represents a novel adaptation.
- The paper supports its claims through a well-designed series of experiments on models like LLaMA2 and LLaMA3, benchmarked against multiple datasets (Enron and Fraud email datasets) and evaluated using diverse attack methods.
-  The PPA method’s adaptability, allowing control over the degree of forgetting, offers practical value by enabling fine-tuning based on specific privacy requirements.

**Weaknesses:**

-  The study primarily focuses on phone numbers and physical addresses as examples of PII. While this provides a starting point, PII often includes more complex and variable types, such as emails, social security numbers, or unique identifiers that may be more difficult to detect and selectively forget.
-  The effectiveness depends on accurately identifying "key elements" within PII sequences. However, the method lacks a detailed discussion on the robustness of this sensitivity analysis, especially for more nuanced or less structured PII.
-  Implementing selective forgetting and memory implanting on large-scale models (e.g., with billions of parameters) can be computationally intensive. The paper does not address potential scalability challenges or discuss optimization strategies for efficiently applying PPA to larger models.
- As a last and perhaps pedantic point, the table captions are lacking. There are too many tables, and their captions and discussion are tedious at best. In general, the metrics are not described well, and there are columns such as risk score that list values, and these are not clear. Columns could have down or up arrows to indicate if higher is better or the reverse.
- Figures are not of high quality. For example, figure 3 is not legible on paper.

**Questions:**

- How scalable is this approach?
- How are the values in Table 1 decided?

---

> ### Author Response · Authors · 2024-11-22
> **Responses to Reviewer J1tm**
>
> Dear Reviewer,
>
> We sincerely thank you for your valuable feedback and thoughtful acknowledgment of our work. Below, we present our responses to your comments and questions.
>
> >#### _**More evaluation on different PII.**_
>
> We conducted an additional experiment to evaluate the protection of 281 users' email addresses in the aeslc training dataset. Using Levenshtein distance [1], we compared the predicted email addresses to the ground truth. As shown in the table, PPA successfully defends all users' email addresses against probing attacks while maintaining model performance comparable to other baseline defense methods. *Detailed results can be found in Appendix K*.
>
>  |**Email Defense Model**|**Risk Score**|**Exact Match Score**|**GPT-4o Email Score**|
> |-|-|-|-|
> |**Empty Response**|47.2|40.5|5.1|
> |**Error Injection**|19.6|17.0|5.3|
> |**Unlearning**|1.0|1.0|1.6|
> |**DEPN**|1.0|1.0|1.3|
> |**PPA**|0.0|0.0|5.0|
>
> >#### _**Discuss optimization strategies for PPA.**_
>
> Thank you for raising this concern. Our initial research was conducted under the assumption of abundant resources, providing a controlled environment to validate our approach. We appreciate your suggestion to combine scalability and optimization strategies for PPA, which will be invaluable for future work.
>
> We have also conducted an initial investigation into the scalability and optimization strategies for PPA. Our experiments involved combining PPA with efficient fine-tuning techniques, such as LoRA [2], using a rank of 16 and an alpha value of 32. As shown in the table below, applying LoRA to PPA produced promising results: after fine-tuning for three epochs, the risk score reduced to 1.0, and after four epochs, it further decreased to 0.0, all while maintaining comparable model performance. Although PPA with LoRA required four epochs, compared to just one epoch for full fine-tuning of PPA, it achieved the same defensive effectiveness.
>
> |**Phone Defense Model**|**Risk Score**|**Exact Match Score**|**GPT-4o Email Score**|
> |-|-|-|-|
> |**PPA LoRA 1-epoch**|24.6|23.6|5.4|
> |**PPA LoRA 2-epoch**|5.4|5.2|5.2|
> |**PPA LoRA 3-epoch**|1.0|1.0|5.0|
> |**PPA LoRA 4-epoch**|0.0|0.0|5.1|
>
> These results demonstrate that PPA has potential for scalability. Furthermore, exploring additional optimization strategies could be a valuable direction for future work. We sincerely thank you for this valuable suggestion. *Detailed results can be found in Appendix M*.
>
> >#### _**Enhances the paper's presentation.**_
>
> Thank you very much for your feedback. We will carefully consider your suggestions and incorporate them into our paper. We will also make the necessary adjustments to the figures.
>
> >#### _**Address Risk Score Table meaning.**_
>
>  |**Category**|**Address Risk Score**|
> |-|-|
> |**Country**|0.005|
> |**Region**|0.1|
> |**SubRegion**|0.15|
> |**Municipality**|0.2|
> |**PostalCode**|0.3|
> |**Street**|0.3|
> |**AddressNumber**|0.3|
>
> Appreciate your query. We designed Table 1 to assess the relative risk of exposing users' physical addresses by category. For example, we consider a street address to be 60 times more sensitive than a country. Specifically, the risk value for a country is 0.005, while for a street it is 0.3.
> The ratio of street to country risk is calculated as 0.3 / 0.005 = 60. The Address Risk Score Table helps identify the exposure risk associated with users' physical addresses.
>
> Reference:
>
> [1] Po, Daw Khin. "Similarity based information retrieval using Levenshtein distance algorithm." _Int. J. Adv. Sci. Res. Eng_ 6.04 (2020): 06-10.
>
> [2] Hu, Edward J., et al. "Lora: Low-rank adaptation of large language models." _arXiv preprint arXiv:2106.09685_ (2021).
>
> **Thank you once again for your valuable feedback and continued support for our work! We sincerely appreciate these constructive and meaningful discussions, as they contribute to making our work more thorough and robust. Should you have any further questions, we would be more than happy to address and discuss them!**
>
> Sincerely
>
> Authors

---

> > ### Comment · Reviewer_J1tm · 2024-11-29
> >
> > Thank you very much for your responses. My rating is already in the accept range, and I want to keep it.

---

> > > ### Author Response · Authors · 2024-11-30
> > >
> > > Dear Reviewer,
> > >
> > > Thank you for taking the time to review our rebuttal and for your encouraging feedback. We are pleased to hear that we have effectively addressed your concerns. Your thoughtful insights are greatly appreciated, and we are committed to further improving our work based on your constructive suggestions.
> > >
> > > Once again, we sincerely thank you for your time and effort.
> > >
> > > Best regards,
> > >
> > > The Authors

---

### Official Review · Reviewer_iQMj · 2024-11-04

**Soundness:** 3
**Presentation:** 3
**Contribution:** 3
**Rating:** 8
**Confidence:** 3

**Summary:**

This work studies the unlearning of personally identifiable information (PII) in large language models (LLMs), proposing a three-step approach called Proactive Privacy Amnesia (PPA). Given target PIIs to forget, the approach first identifies the key elements in memorized PII, then focuses exclusively on forgetting those key elements, and finally implants alternative information.

**Strengths:**

1. The problem of unlearning in LLMs is critical for conforming to the "right to be forgotten" legal principle and is essential for protecting personal information from misuse by LLM producers.

2. The paper introduces the concept of the "memorization factor" to quantify the sensitivity of PII.

**Weaknesses:**

Given the numerous unlearning targets, the resulting unlearned model may degrade in various aspects. While current evaluation metrics for LLM performance include perplexity and email completion, additional metrics are needed to measure the utility loss due to unlearning. Specific metrics from the paper "Knowledge Unlearning for Mitigating Privacy Risks in Language Models" could be particularly relevant. Consider including metrics such as TruthfulQA and HellaSwag.

**Questions:**

1. In some unlearning tasks, such as Physical Address Defense in Table 4, PPA does not appear to outperform DEPN. A more detailed analysis of the cases where PPA underperforms/outperforms DEPN is warranted.

2. A question arises regarding the security of the unlearning paradigm. The unlearning of specific PII is reasonable if the PII was indeed learned by the LLM. How can individuals or LLM producers verify that the PII has been learned? Additionally, is the proposed concept of the "memorization factor" robust enough to detect unreasonable PII unlearning queries?

---

> ### Author Response · Authors · 2024-11-22
> **Responses to Reviewer iQMj**
>
> Dear Reviewer,
>
> We deeply appreciate your insightful feedback and recognition of various aspects of our work. Our responses to your comments and questions are provided below.
>
> >#### _**Additional utility metrics.**_
>
> Thank you for your valuable feedback. We provide some new evaluations on the model's performance metrics on both MMLU [1] and TruthfulQA [2]. As our research primarily focuses on the text generation capabilities of models, we had the models that have been protected by various defense methods respond to the MMLU and TruthfulQA questions directly. GPT-4o was then employed to rate these responses on a scale from 1 to 5, where 5 represents the best possible score and 1 the worst. Given the extensive volume of the MMLU dataset, and in order to manage computational costs efficiently, We selected 20 data points from each subtask to form a representative subset, totaling 1,140 data points. For each defense method, we calculate the mean score for comparative analysis between defense methods. We found that PPA achieves the highest MMLU and TruthfulQA score among all baseline defense methods. *Detailed results can be found in Appendix F.*
>
> |**Phone Defense Model**|**MMLU Score**|**TruthfulQA Score**|**GPT-4o Email Score**|
> |-|-|-|-|
> |**Empty Response**|3.3|3.4|5.7|
> |**Error Injection**|3.3|3.2|5.2|
> |**Unlearning**|1.6|1.7|1.1|
> |**DEPN**|2.3|2.4|2.0|
> |**PPA**|4.2|4.1|5.2|
>
> >#### _**More detailed analysis between PPA and DEPN.**_
>
> We truly appreciate your insightful feedback. Table 4 is based on the Fraud Email dataset, where the physical addresses are simpler compared to those in the Enron Email dataset. For example, an address in the Fraud Email dataset, "Brian Smith address London", whereas in the Enron Email dataset, "Jeffrey Dasovich address 101 California St. Suite 1950". This comparison highlights that PPA is better equipped to handle more complex physical addresses, demonstrating that PPA is a more generalizable method compared to DEPN.
>
> >#### _**How to verify that LLMs have learned PII.**_
>
> The unlearning of specific PII is reasonable if the PII was indeed learned by the LLM. In fact, our ability to successfully attack the model indicates that the LLM has learned the PII. For example, our attacks achieving an exact match score demonstrate that the LLM has fully memorized it. Specifically, we found that with no defense on phone numbers, the exact match score reached 56.0.
>
> >#### _**More evaluation on different PII.**_
>
> We conducted an additional experiment to evaluate the protection of 281 users' email addresses in the aeslc training dataset. Using Levenshtein distance [3], we compared the predicted email addresses to the ground truth. As shown in the table, PPA successfully defends all users' email addresses against probing attacks while maintaining model performance comparable to other baseline defense methods. These results demonstrate that PPA effectively safeguards users' PII. *Detailed results can be found in Appendix K*.
>
>  |**Email Defense Model**|**Risk Score**|**Exact Match Score**|**GPT-4o Email Score**|
> |-|-|-|-|
> |**Empty Response**|47.2|40.5|5.1|
> |**Error Injection**|19.6|17.0|5.3|
> |**Unlearning**|1.0|1.0|1.6|
> |**DEPN**|1.0|1.0|1.3|
> |**PPA**|0.0|0.0|5.0|
>
> Reference:
>
> [1] Hendrycks, Dan, et al. "Measuring massive multitask language understanding." _arXiv preprint arXiv:2009.03300_ (2020).
>
> [2] Lin, Stephanie, Jacob Hilton, and Owain Evans. "Truthfulqa: Measuring how models mimic human falsehoods." _arXiv preprint arXiv:2109.07958_ (2021).
>
> [3] Po, Daw Khin. "Similarity based information retrieval using Levenshtein distance algorithm." _Int. J. Adv. Sci. Res. Eng_ 6.04 (2020): 06-10.
>
> **Thank you once again for your insightful feedback and support for our work! We deeply appreciate these meaningful and constructive discussions, as they help make our work more thorough and robust. If you have any additional questions, we would be delighted to discuss and address them!**
>
> Sincerely,
>
> Authors

---

> > ### Comment · Reviewer_iQMj · 2024-11-26
> >
> > Thanks for the reponse and the additional results. I will increase my score.

---

> > > ### Author Response · Authors · 2024-11-26
> > >
> > > Dear Reviewer,
> > >
> > > Thank you for reviewing our rebuttal and for your encouraging feedback. We are delighted to know that we have successfully addressed your concerns. Your thoughtful insights are deeply valued, and we remain committed to enhancing our work based on your constructive suggestions.
> > >
> > > Once again, we sincerely appreciate your time and effort.
> > >
> > > Best regards,
> > >
> > > The Authors

---

### Official Review · Reviewer_Pc5A · 2024-11-04

**Soundness:** 3
**Presentation:** 3
**Contribution:** 2
**Rating:** 6
**Confidence:** 3

**Summary:**

In this paper, the authors propose a scheme to selectively protect PII in Large Language Model training data. The approach consists of three major components: sensitivity analysis, selective forgetting, and memory implantation. The results are evaluated across multiple LLM attacks, demonstrating that the defense shows promising effectiveness.

**Strengths:**

1. Has theoretical justification for methods. It's impressive that the authors clearly explain the method's concept with theoretical justification, which definitely helps readers understand.
2. Includes multiple attacks for evaluation. The authors consider multiple attacks in the evaluation, showcasing that the proposed method is robust under different settings.
3. Results seem promising with utility-privacy tradeoff. Based on the results, the proposed method appears promising, though slightly less effective than some prior work in certain scenarios. The authors explain the advantages of the proposed solution well.

**Weaknesses:**

1. Method seems straightforward; each component seems lacks novelty. The sensitivity analysis appears to follow standard definitions of PII in language models. The selective forgetting component proposes a loss function, but it is relatively simple, and the memory implanting seems directly referenced from prior works. Additional insights and modifications to tailor these components specifically for PII protection may be needed to make the method more innovative. For example, consider designing the memory implanting to optimize performance for different types of PII. Similar innovations should be highlighted for each component.
2. Evaluation may benefit from additional metrics, such as exposure. I feel that including more metrics beyond attack metrics and model performance metrics would provide more comprehensive insights into the proposed method. For instance, adding an exposure metric, as discussed in [1], could measure the memorization and likelihood of extraction when partial information is not protected. This evaluation would also better align with the design of other baselines compared in this work and help readers better understand the effectiveness of the proposed method.
3. No comparison with differential privacy-based methods. There seems to be a lack of comparison with differential privacy-based methods in the evaluation, such as [2]. Since differential privacy is a mainstream defense approach, comparing it with selective-DP should also be considered in this paper. More discussion and comparisons on the privacy guarantee and utility to highlight the pros and cons of the proposed method and the dp-based methods will be appreciated.

[1] Carlini, Nicholas, et al. "The secret sharer: Evaluating and testing unintended memorization in neural networks." 28th USENIX security symposium (USENIX security 19). 2019.

[2] Shi, Weiyan, et al. "Just Fine-tune Twice: Selective Differential Privacy for Large Language Models." Proceedings of the 2022 Conference on Empirical Methods in Natural Language Processing. 2022.

**Questions:**

1. Will the method leak any other user's PII when it's not protecting all samples, causing some originally safe information to be exposed?

---

> ### Author Response · Authors · 2024-11-22
> **Responses to Reviewer Pc5A (Part 1/2)**
>
> Dear Reviewer,
>
> We greatly value your thoughtful feedback and acknowledgment of different aspects of our work. We are thankful to have the chance to address your questions and concerns.
>
> >#### _**Novelty**_
>
> Firstly, we thank you for your suggestions. We acknowledge that novelty is not our primary contribution; our approach is simple and practical, aimed at prioritizing more stable and reliable implementations for privacy protection. **We are grateful for your suggestions to improve our memory implanting technique, and have made some attempts, such as integrated selective analysis into the memory implanting process.** In a supplementary experiment, we modified the memory implanting component to focus on replacing the key element with a different token. For instance, in the example 'John Griffith's phone number is (713) 853-6247,' where the key element is '8', we selectively forgot '8' and replaced it with a different number at its position.
>
> As shown in the table, we observe that the Modified Memory Implanting PPA provides the same protection for users' phone numbers and outperforms PPA in GPT-4o EmailScore by approximately 9.6%.
>
>  |**Phone Defense Model**|**Risk score**|**Exact Match Score**|**GPT-4o EmailScore**|
> |-|-|-|-|
> |**Empty Response**|37.2|34.8|5.7|
> |**Error Injection**|19.3|17.6|5.2|
> |**Unlearning**|0.0|0.0|1.1|
> |**DEPN**|0.0|0.0|2.0|
> |**PPA**|0.0|0.0|5.2|
> |**Modified Memory Implanting PPA**|0.0|0.0|5.7|
>
> Address substitution presents challenges because addresses are highly contextually dependent. Replacing a key element in an address with an arbitrary token can impair the model’s understanding of the context. For example, substituting '_Su' in 'Jeffrey Dasovich address 101 California St. Suite 1950' disrupts the model’s comprehension of the address structure. Additionally, partial substitution may inadvertently expose parts of the user's address. Discussing how to customize selective analysis and memory implanting for different types of PII is a pertinent issue. Your suggestion to design memory implanting to optimize performance for various PII types is valuable. **Given that our current priority is to ensure stable and reliable privacy protection, we continue to opt for PPA. We will prioritize your suggestions as key components of our future work. Thank you again for your invaluable feedback.** *Detailed results can be found in Appendix I.*
>
>
> >#### _**Adding the Exposure metric, as discussed in [1]**_
>
> Thank you for your suggestion. The exposure metric [1] is an effective method for detecting whether an model has memorized PII. We applied the exposure metric across all baseline methods for our evaluation. We calculate the exposure metric using all baseline methods. Calculating the exposure of PII is time-consuming; therefore, referencing Table 2 as outlined in The Secret Sharer [1], we calculated the exposure for 10 phone numbers. The average exposure across these numbers is presented in the table below.
>
> As the table illustrates, PPA outperforms other baseline defense methods. *Detailed results can be found in Appendix J.*
>
> |**Phone Defense Model**|**Exposure**|
> |-|-|
> |**Empty Response**|12.50|
> |**Error Injection**|10.94|
> |**Unlearning**|3.55|
> |**DEPN**|7.72|
> |**PPA**|0.05|
>
>
> >#### _**Comparison with differential privacy-based methods**_
>
> Thank you for your insightful feedback. We implemented Just Fine-Tune Twice (JFT) [2] and Differentially Private Decoding (DP Decoding) [3]. To evaluate these methods, we conducted probing attacks on both JFT and DP Decoding. Specifically, for DP Decoding, we tested various values of the lambda parameter ranging from 0.1 to 0.9 and selected a result where the utility score was worse than that of PPA. We observed that our PPA method outperformed both DP Decoding and JFT, achieving a lower risk score and a higher utility score. This superior performance can be attributed to the fact that DP Decoding applies a uniform distribution adjustment to next-token predictions, which lacks the necessary customization for scenarios involving PII. *Detailed results and discussion can be found in Appendix E.*
>
> |**Phone Defense Model**|**Risk Score**|**Exact Match Score**|**GPT-4o Email Score**|
> |-|-|-|-|
> |**JFT [2]**|28.4|26.0|5.0|
> |**DP Decoding [3]**|30.4|28.4|4.7|
> |**PPA**|0.0|0.0|5.2|
>
> Reference:
>
> [1] Carlini, Nicholas, et al. "The secret sharer: Evaluating and testing unintended memorization in neural networks." 28th USENIX security symposium (USENIX security 19). 2019.
>
> [2]  Shi, Weiyan, et al. "Just Fine-tune Twice: Selective Differential Privacy for Large Language Models." Proceedings of the 2022 Conference on Empirical Methods in Natural Language Processing. 2022.
>
> [3] Majmudar, Jimit, et al. "Differentially private decoding in large language models." _arXiv preprint arXiv:2205.13621_ (2022).

---

> > ### Author Response · Authors · 2024-11-24
> > **Responses to Reviewer Pc5A (Part 2/2)**
> >
> > >#### _**Originally safe information to be exposed?**_
> >
> > Thank you for your suggestion. While we may not have fully understood your point, we interpreted your concern as questioning whether the PPA defense might inadvertently expose previously secure PII of users who are not directly protected by the PPA method. Following your excellent suggestion, we utilized the exposure metric [1] to detect whether originally secure information becomes exposed. Based on this interpretation, we conducted the following experiment:
> >
> > We evaluated the exposure metric for safe phone numbers—those not exposed to attackers—that were not protected by the PPA method, using both the no-defense setup and the PPA model. Given the time-intensive nature of calculating PII exposure, we referenced Table 2 from The Secret Sharer [1] and analyzed the exposure of 10 such phone numbers. The average exposure for these cases is summarized in the table below.
> >
> > As shown in the table, the exposure of phone numbers not protected by the PPA method decreases slightly, from 1.57 (no defense) to 1.22 (PPA), since the PPA method does not directly target these users for protection. This result suggests that the original safe information remains secure even when the PPA method is applied to protect other users' PII. *Detailed results and discussion can be found in Appendix L.*
> >
> > |**Phone Defense Model**|**Exposure**|
> > |-|-|
> > |**No Defense**|1.57|
> > |**PPA**|1.22|
> >
> > Reference:
> >
> > [1] Carlini, Nicholas, et al. "The secret sharer: Evaluating and testing unintended memorization in neural networks." 28th USENIX security symposium (USENIX security 19). 2019.
> >
> > **Thank you once again for your thoughtful insights and support for our work! We greatly value these constructive and meaningful discussions with you, as they contribute to making our work more comprehensive and robust. If you have any further questions, we would be more than happy to discuss and address them!**
> >
> > Sincerely,
> >
> > Authors

---

> ### Comment · Reviewer_Pc5A · 2024-11-24
>
> Thank you for your responses. Could you please provide more details about the parameter selection for the added DP experiment and explain why it is reasonable and comparable to the proposed method? The current description: "we tested various values of the lambda parameter ranging from 0.1 to 0.9 and selected a result where the utility score was worse than that of PPA"  is a bit vague. Including this detailed information in the paper would also be beneficial.

---

> ### Author Response · Authors · 2024-11-24
>
> Thank you for your thoughtful questions. Because DP Decoding [1] aims to protect users' PII in an already trained model, we added DP Decoding [1] into our experiment. DP Decoding employs a computationally lightweight perturbation mechanism at the decoding stage. The core principle of this method is illustrated by the equation q' = λq + (1 − λ)u, where q' represents the perturbed distribution—a linear interpolation between the original distribution q and the uniform distribution u. The parameter λ impacts the utility (the effectiveness of the generated output) and privacy (the protection of sensitive information). As λ varies from 0 to 1, a value closer to 0 enhances privacy but reduces utility, whereas a value closer to 1 improves utility at the expense of privacy.
> Therefore, we selected results for DP Decoding where λ ranged from (0.1, 0.3, 0.5, 0.7, 0.9), as illustrated in the table below.
>
> ||**PPA**|**λ=0.1**|**λ=0.3**|**λ=0.5**|**λ=0.7**|**λ=0.9**|**no defense**|
> |-|-|-|-|-|-|-|-|
> |**Risk Score**|0.0|15.7|**30.4**|32.4|33.2|41.4|57.9|
> |**Exact Match Score**|0.0|10.6|**28.4**|30.1|31.1|38.6|56.0|
> |**GPT-4o Email Score**|5.2|3.0 (3.025)|**4.8 (4.750)**|4.8 (4.825)|4.9 (4.875)|5.1 (5.125)|5.0|
>
> In the table, we observe a trend across λ values of 0.1, 0.3, 0.5, 0.7, to 0.9, showing a progression from low to high risk scores and from low to high utility. We selected λ=0.3 as it achieved the best balance between utility and privacy protection among the tested values, offering better utility compared to λ=0.1, and improved defense capability compared to λ=0.5.
>
> These results demonstrated that the utility score was lower than that of PPA and that PPA's defense capability was superior to that of DP Decoding, thereby illustrating that PPA outperforms DP Decoding. *Detailed results can be found in Appendix E*. (We apologize for the typo in our previous results. According to the rounding rules, λ=0.3 should be 4.8, not 4.7.)
>
>
> **Thank you for your time and effort. We hope our responses address your concerns. If you have any further questions, we will do our best to provide clarification. We will also incorporate all your suggestions and our revisions into the paper. Thank you very much for helping make our paper more complete.**
>
>
> Reference:
>
> [1] Majmudar, Jimit, et al. "Differentially private decoding in large language models." _arXiv preprint arXiv:2205.13621_ (2022).

---

> > ### Comment · Reviewer_Pc5A · 2024-11-26
> >
> > Thanks for your response, I have raise the score

---

> > > ### Author Response · Authors · 2024-11-26
> > >
> > > Dear Reviewer,
> > >
> > > Thank you for reviewing our rebuttal and for your positive feedback. We’re glad to hear that we’ve addressed your concerns effectively. Your valuable insights are greatly appreciated, and we are committed to refining our work further based on your suggestions.
> > >
> > > Thank you again for your time and thoughtful effort.
> > >
> > > Best regards,
> > >
> > > The Authors

---

### Official Review · Reviewer_Da8V · 2024-11-04

**Soundness:** 3
**Presentation:** 2
**Contribution:** 3
**Rating:** 8
**Confidence:** 3

**Summary:**

The paper proposes Proactive Privacy Amnesia, to unlearn PII in LLM while preserving its utility. The framework is composed of sensitivity analysis, selective forgetting, and memory implanting. The empirical results demonstrate the effectiveness in eliminating the privacy leakage risk.

**Strengths:**

- The sensitivity analysis is interesting and inspiring, minimizing the impact on model performance.
- The proposed framework outperforms the baselines in terms of utility and defense ability.

**Weaknesses:**

- The authors can compare their framework with differentially private decoding [1] and finetuning [2] methods, which also prevent the model from outputing sensitive information.

- Limited evaluation on model utility. The author only evaluate the model's utility on the same unlearning dataset, i.e., Enron Email. However, it's desirable to evaluate the model's performance on general downstream tasks, such as the GLUE and MMLU dataset.

- The sensitivity analysis aims to isolate tokens that carry a higher amount of information. What if the attacker has some prior knowledge of the phone number/address? It may increase the attack success rate by conditioning the prediction on the former part of the private information. Furthermore, would unlearning the latter part, i.e., the tokens after the top-k, lead to better performance?

[1] Majmudar, J., Dupuy, C., Peris, C., Smaili, S., Gupta, R., & Zemel, R. (2022). Differentially private decoding in large language models. arXiv preprint arXiv:2205.13621.

[2] Li, Xuechen, Florian Tramer, Percy Liang, and Tatsunori Hashimoto. "Large Language Models Can Be Strong Differentially Private Learners." In International Conference on Learning Representations.

**Questions:**

- The sensitivity analysis is expected to reduce the impact on utility of the LLM. However, in Table 5, Unlearning + Memory Implanting shows model similar performance with Proactive Privacy Amnesia, with increased attack success rate. How to interprete such result? Why the sensitivity analysis could decrease the attack score?

- What's the average proportion of the top-k token in terms of the total length of the private information?

---

> ### Author Response · Authors · 2024-11-22
> **Responses to Reviewer Da8V (Part 1/2)**
>
> Dear Reviewer,
>
> We sincerely appreciate your constructive feedback and recognition of various aspects of our work. Below, we provide responses to your comments and questions.
>
> >#### _**Compare with additional methods**_
>
> Thank you for your insightful feedback. We implemented the Differentially Private Decoding (DP Decoding) in [1] and the Just Fine-Tune Twice (JFT) method in [3], building on the framework introduced in [2]. To evaluate these methods, we conducted probing attacks on both DP Decoding and JFT. Specifically, for DP Decoding, we tested various values of the lambda parameter ranging from 0.1 to 0.9 and selected the result that achieved the best balance between utility and privacy protection. We observed that our PPA method still outperformed both DP Decoding and JFT, achieving a lower risk score and a higher utility score. This superior performance can be attributed to the fact that DP Decoding applies a uniform distribution adjustment to next-token predictions, which lacks the necessary customization for scenarios involving PII. *Detailed results can be found in Appendix E.*
>
> |**Phone Defense Model**|**Risk Score**|**Exact Match Score**|**GPT-4o Email Score**|
> |-|-|-|-|
> |**DP Decoding[1]**|30.4|28.4|4.8|
> |**JFT[3]**|28.4|26.0|5.0|
> |**PPA**|0.0|0.0|5.2|
>
>
> >#### _**More evaluation on model utility.**_
>
> Thank you for your valuable feedback. We provide some new evaluations on the model's performance metrics on both MMLU [4] and TruthfulQA [5]. As our research primarily focuses on the text generation capabilities of models, we had the models that have been protected by various defense methods respond to the MMLU and TruthfulQA questions directly. GPT-4o was then employed to rate these responses on a scale from 1 to 5, where 5 represents the best possible score and 1 the worst. Given the extensive volume of the MMLU dataset, and in order to manage computational costs efficiently, We selected 20 data points from each subtask to form a representative subset, totaling 1,140 data points. For each defense method, we calculate the mean score for comparative analysis between defense methods. We found that PPA achieves the highest MMLU and TruthfulQA score among all baseline defense methods. *Detailed results can be found in Appendix F.*
>
> |**Phone Defense Model**|**MMLU Score**|**TruthfulQA Score**|**GPT-4o Email Score**|
> |-|-|-|-|
> |**Empty Response**|3.3|3.4|5.7|
> |**Error Injection**|3.3|3.2|5.2|
> |**Unlearning**|1.6|1.7|1.1|
> |**DEPN**|2.3|2.4|2.0|
> |**PPA**|4.2|4.1|5.2|
>
> In our paper, we evaluate the model's utility not only on the same unlearning dataset but also using an email completion metric, which assesses the model's performance on additional truncated emails, with GPT-4o judging the completion email scores.
>
> >#### _**Assume Attacker has prior knowledge of the PII**_
>
> Appreciate your query. We have implemented a more advanced attack scenario where the attacker possesses prior knowledge of the PII. This is, in fact, a more challenging scenario, and we are more than willing to explore it further. Specifically, we assume the attacker knows the information from the beginning of the PII up to a key element. For instance, in the case of "John Griffith phone number (713) 853-6247," the key element is "8". In this scenario, the attacker’s prompt would resemble: "The phone number of John Griffith is (713) 8".
>
> As shown in the table, we observe that PPA achieves the best balance between defense capability and model performance. *Detailed results can be found in Appendix G.*
>
>  |**Phone Defense Model**|**Risk score**|**Exact Match Score**|**GPT-4o EmailScore**|
> |-|-|-|-|
> |**Empty Response**|154.0|141.5|5.7|
> |**Error Injection**|75.1|69.1|5.2|
> |**Unlearning**|6.7|1.8|1.1|
> |**DEPN**|36.2|27.3|2.0|
> |**PPA**|12.1|9.8|5.2|
>
> Reference:
>
> [1] Majmudar, J., Dupuy, C., Peris, C., Smaili, S., Gupta, R., & Zemel, R. (2022). Differentially private decoding in large language models. arXiv preprint arXiv:2205.13621.
>
> [2] Li, Xuechen, Florian Tramer, Percy Liang, and Tatsunori Hashimoto. "Large Language Models Can Be Strong Differentially Private Learners." In International Conference on Learning Representations.
>
> [3] Shi, W., Shea, R., Chen, S., Zhang, C., Jia, R., & Yu, Z. (2022). Just fine-tune twice: Selective differential privacy for large language models. _arXiv preprint arXiv:2204.07667_.
>
> [4] Hendrycks, Dan, et al. "Measuring massive multitask language understanding." _arXiv preprint arXiv:2009.03300_ (2020).
>
> [5] Lin, Stephanie, Jacob Hilton, and Owain Evans. "Truthfulqa: Measuring how models mimic human falsehoods." _arXiv preprint arXiv:2109.07958_ (2021).

---

> ### Author Response · Authors · 2024-11-22
> **Responses to Reviewer Da8V (Part 2/2)**
>
> >#### _**Explain the relationship between Sensitivity Analysis and Unlearning + Memory Implanting**_
>
> We sincerely appreciate your insightful feedback, and you are correct. The sensitivity analysis is expected to lessen the impact on the utility of the LLM. As shown in Table 6, the combination of unlearning and memory implanting results in poorer performance compared to PPA. However, it also demonstrates better defense capabilities.
>
>
>  |**Table 6 - Physical Address Ablation Study**|**Model Performance**|**Risk Score**|
> |-|-|-|
> |**Proactive Privacy Amnesia**|19.5|7.3|
> |**Unlearning + Memory Implanting**|33.6|6.7|
>
> The reason why phone numbers deviate from expectations, as shown in Table 5, is due to their simple format, such as (xxx) xxx-xxxx. Consequently, memory implanting more effectively restores utility, leading to the observed similarity in performance between Unlearning + Memory Implanting and Proactive Privacy Amnesia. Additionally, due to the simplicity of the phone number format, Sensitivity Analysis proves sufficient for defense.
>
>  |**Table 5 - Phone Number Ablation Study**|**Model Performance**|**Risk Score**|
> |-|-|-|
> |**Proactive Privacy Amnesia**|16.0|0.0|
> |**Unlearning + Memory Implanting**|16.3|0.2|
>
>
> >#### _**Average proportion of the key element relative to the total length of the PII.**_
>
> The proportions of phone numbers and physical addresses are 6.7% and 27.6%, respectively. *Detailed results can be found in Appendix H.*
>
> **We sincerely thank you for your thoughtful insights and support for our work! Engaging in discussions about these constructive and meaningful questions has been truly rewarding, as it helps make our work more comprehensive and robust. If you have any additional questions, we would be delighted to discuss and address them!**
>
> Best regards,
>
> Authors

---

> ### Comment · Reviewer_Da8V · 2024-11-25
> **Reply to Authors' Rebuttal**
>
> I appreciate the authors' significant effort during the rebuttal. I still have one question for the sensitivity analysis:
>
> > Furthermore, would unlearning the latter part, i.e., the tokens after the top-k, lead to better performance?
>
> I'm curious if there's any difference between the cases: 1) unlearn only the top-k tokens; 2) unlearn some middle parts that span the k-th token; 3) unlearn some tokens from the k-the token. Why do the authors choose (1) in the paper?

---

> > ### Author Response · Authors · 2024-11-25
> >
> > Thank you for your question, and apologies for not addressing this earlier. It seems the question is a bit unclear, so we would like to clarify the following points:
> >
> > 1.  **unlearn only the top-k tokens**: This refers to forgetting the most sensitive token.
> > 2.  **unlearn some middle parts that span the k-th token**: This involves forgetting the most sensitive token along with one token before and one token after it.
> > 3.  **unlearn some tokens from the k-the token**: This means forgetting the most sensitive token as well as all the tokens that follow it.
> >
> > We greatly appreciate your inquiry and are more than happy to provide clarification.

---

> > > ### Comment · Reviewer_Da8V · 2024-11-25
> > >
> > > Thanks for your response. How do the three strategies impact the model performance & unlearning effect? Why do you select the first strategy in your paper?

---

> > > > ### Author Response · Authors · 2024-11-25
> > > >
> > > > Thank you so much for your clarification. We will further investigate the three points mentioned above and provide more discussion soon. We sincerely appreciate your help in making our paper more comprehensive.

---

> > > > > ### Author Response · Authors · 2024-11-25
> > > > >
> > > > > Dear Reviewer,
> > > > >
> > > > > We sincerely appreciate your constructive feedback, which has significantly contributed to improving our paper's clarity and comprehensiveness. We conducted some ablation stuides on protecting the user's physical address using three following strategies, as outlined below, with their respective Risk Score and GPT-4o Email Score presented in the table:
> > > > >
> > > > > 1. **Only the most sensitive tokens (PPA)**
> > > > > 2. **Span around the most sensitive token**
> > > > > 3. **The most sensitive tokens + the following tokens**
> > > > >
> > > > > Here is the comparison of the strategies:
> > > > >
> > > > > | **Forgetting Strategies**                         | **Risk Score** | **GPT-4o Email Score** |
> > > > > |-------------------------------------------------------|----------------|------------------------|
> > > > > | **Only the most sensitive tokens (PPA)**              | 4.7 (4.702)    | 3.6                    |
> > > > > | **Span around the most sensitive token**              | 4.7 (4.737)    | 3.4                    |
> > > > > | **The most sensitive tokens + the following tokens**  | 4.4            | 2.9                    |
> > > > >
> > > > > We find that:
> > > > > - **"Only the most sensitive token (PPA)"** achieves an optimal balance between defense capability and model performance.
> > > > > - **"Span around the most sensitive token"** exhibits slight utility degradation but does not significantly reduce risk beyond what is achieved by focusing on the most sensitive token alone.
> > > > > - **"The most sensitive tokens + the following tokens"** strategy reduces the risk score further but at the cost of significantly lowering the GPT-4o Email Score. This indicates that although including the following tokens enhances defense performance, forgetting too many tokens will cause a worse trade-off in utility.
> > > > >
> > > > > Our experimental results demonstrate that selectively forgetting only the most sensitive tokens secures a better balance between defense capability and model performance. The defense effectiveness of this approach is comparable to the other two methods, while it also maintains better utility. Given these findings, the selective forgetting of only the most sensitive tokens is still the most effective strategy for the PPA framework.
> > > > >
> > > > > **Once again, we sincerely thank you for your constructive feedback and valuable suggestions, which have greatly contributed to improving the quality and clarity of our paper. We will incorporate the experiments you proposed as new ablation studies in our revised manuscript. Detailed results can be found in Appendix N.**
> > > > >
> > > > > Best regards,
> > > > >
> > > > > Authors

---

> > > > > > ### Comment · Reviewer_Da8V · 2024-11-26
> > > > > >
> > > > > > Thanks for the response! I have updated the score. Good luck:)

---

> > > > > > > ### Author Response · Authors · 2024-11-26
> > > > > > >
> > > > > > > Dear Reviewer,
> > > > > > >
> > > > > > > Thank you very much for taking the time to review our rebuttal and for your positive feedback. We are pleased to hear that we have addressed your concerns. Your insightful feedback is greatly appreciated, and we are committed to continuously improving our work based on your valuable suggestions.
> > > > > > >
> > > > > > > Thank you once again for your dedicated time and effort.
> > > > > > >
> > > > > > > Sincerely,
> > > > > > >
> > > > > > > Authors

---

### Author Response · Authors · 2024-11-22
**General Statement**

Dear Reviewers,

We sincerely thank you for your valuable comments and suggestions. We appreciate the opportunity to address your concerns and improve our manuscript. The changes have been highlighted in blue for text and orange for tables and figures. Below, we outline the major revisions we have made:

**1. Adding Comparison with Differential Privacy-Based Methods.** Based on the reviewers' suggestions, we compare our method with "Differentially private decoding in large language models" (DP Decoding) [1] and "Just fine-tune twice: Selective differential privacy for large language models" (JFT) [2].

To evaluate these methods, we applied probing attacks to both DP Decoding and JFT. We observed that our PPA method outperformed both DP Decoding and JFT, achieving a lower risk score and a higher utility score. *Detailed results and discussion can be found in Appendix E.*


**2. Adding additional model utility evaluation on TruthfulQA [5] and MMLU [6].** Based on the reviewers' suggestions, we evaluated the model's performance metrics on both TruthfulQA and MMLU. We employed GPT-4o to assess the responses from all baseline defense methods. Our findings indicate that PPA achieves the highest TruthfulQA and MMLU scores among all baseline defense methods. *Detailed results can be found in Appendix F.*

**3. Adding the Exposure Metric [3].** As recommended by the reviewers, we calculated the exposure metric for all baseline methods. Since calculating the exposure of PII is computationally intensive, we followed the approach in Table 2 of [3] and evaluated the exposure for 10 phone numbers. Our results show that PPA outperforms other baseline defense methods. *Detailed results can be found in Appendix J.*

**4. More evaluation on different PII.** Based on the reviewers' suggestions, we conducted an additional experiment to evaluate the protection of 281 users' email addresses in the aeslc training dataset. Using Levenshtein distance [4], we compared the predicted email addresses to the ground truth. PPA successfully defends all users' email addresses against probing attacks while maintaining model performance comparable to other baseline defense methods. *Detailed results can be found in Appendix K*.

Reference:

[1] Majmudar, J., Dupuy, C., Peris, C., Smaili, S., Gupta, R., & Zemel, R. (2022). Differentially private decoding in large language models. arXiv preprint arXiv:2205.13621.

[2] Shi, W., Shea, R., Chen, S., Zhang, C., Jia, R., & Yu, Z. (2022). Just fine-tune twice: Selective differential privacy for large language models. _arXiv preprint arXiv:2204.07667_.

[3] Carlini, Nicholas, et al. "The secret sharer: Evaluating and testing unintended memorization in neural networks." 28th USENIX security symposium (USENIX security 19). 2019.

[4] Po, Daw Khin. "Similarity based information retrieval using Levenshtein distance algorithm." _Int. J. Adv. Sci. Res. Eng_ 6.04 (2020): 06-10.

[5] Hendrycks, Dan, et al. "Measuring massive multitask language understanding." arXiv preprint arXiv:2009.03300 (2020).

[6] Lin, Stephanie, Jacob Hilton, and Owain Evans. "Truthfulqa: Measuring how models mimic human falsehoods." arXiv preprint arXiv:2109.07958 (2021).

**We sincerely appreciate your thoughtful suggestions, which have encouraged us to think more deeply about our work. We welcome any additional feedback and are more than happy to address any further questions or concerns you may have.**

Best regards,

Authors

---

### Meta-Review · Area_Chair_f2FB · 2024-12-14

**Metareview:**

## Summary of Contributions

This paper proposes a method to unlearn / protect personally identifiable information (PII) in large language models (LLMs). The method--named Proactive Privacy Amnesia (PPA)--is inspired by anterograde amnesia. The method consists of the following steps: (i) identify sensitive token, (ii) selectively forgetting this token and (iii) implanting false memory instead. The authors perform experiments for their method against previous work, in the settings where PIIs are phone number and physical addresses. They demonstrate that the new approach is more effective (in terms of protection and model utility) compared to previously proposed methods.

## Strengths

- Unlearning in LLMs is an important and timely topic as continued rollouts of (many forms of) LLMs can threaten user's privacy.

- The proposed method is novel and innovative, and should lead to further investigation.

- Some steps, such as sensitivity analysis, has theoretical justifications.

## Weaknesses

- For the original submission, the experiments were not as comprehensive as one would have hope (e.g. no comparison with DP-based methods); this has been addressed during the rebuttal. In addition to this, the improvements are not uniform in all settings, e.g. DEPN is comparable / better in some cases.

- Although the combination of steps are innovative, it seems like "selective forgetting" and "memory planting" themselves are not new and have appeared in literature.

- The presentation can be improved. For example, the figures and captions are not as clear as one would hope. The methods--especially the "selective forgetting" and "memory planting" steps--are not described as clear as it could be (e.g. once we the loss function in (6) what do we do next? gradient descent?). Moreover, some notations are not clearly defined (e.g. $p, q$ in Equation (1)).

## Recommendation

Given that the problem is important, the method is innovation and shows promising empirical results, we recommend acceptance.

**Additional Comments On Reviewer Discussion:**

As mentioned in the meta-review, the original version of the paper do not have comprehensive experiments. Most of the rebuttals are spent on this. Namely, the authors have added many experiments to the paper. I think this makes the comparison much more comprehensive.

---

### Decision · Program_Chairs · 2025-01-22

Accept (Poster)